# Latent Graph Inference with Limited Supervision

**Jianglin Lu**[1*]   **Yi Xu**[1]   **Huan Wang**[1]   **Yue Bai**[1]   **Yun Fu**[1,2]
[1]Department of Electrical and Computer Engineering, Northeastern University
[2]Khoury College of Computer Science, Northeastern University
Project Page: https://jianglin954.github.io/LGI-LS/

## Abstract

Latent graph inference (LGI) aims to jointly learn the underlying graph structure and node representations from data features. However, existing LGI methods commonly suffer from the issue of *supervision starvation*, where massive edge weights are learned without semantic supervision and do not contribute to the training loss. Consequently, these supervision-starved weights, which may determine the predictions of testing samples, cannot be semantically optimal, resulting in poor generalization. In this paper, we observe that this issue is actually caused by the graph sparsification operation, which severely destroys the important connections established between pivotal nodes and labeled ones. To address this, we propose to *restore the corrupted affinities and replenish the missed supervision for better LGI*. The key challenge then lies in identifying the critical nodes and recovering the corrupted affinities. We begin by defining the pivotal nodes as *k-hop starved nodes*, which can be identified based on a given adjacency matrix. Considering the high computational burden, we further present a more efficient alternative inspired by *CUR matrix decomposition*. Subsequently, we eliminate the starved nodes by reconstructing the destroyed connections. Extensive experiments on representative benchmarks demonstrate that reducing the starved nodes consistently improves the performance of state-of-the-art LGI methods, especially under extremely limited supervision (6.12% improvement on Pubmed with a labeling rate of only 0.3%).

## 1  Introduction

Graph neural networks (GNNs) [8, 12, 23, 33] have recently received considerable attention due to their strong ability to handle complex graph-structured data. GNNs consider each data sample as a node and model the affinities between nodes as the weights of edges. The edges as a whole constitute the graph structure or topology of the data. By integrating the graph topology into the training process of representation learning, GNNs have achieved remarkable performance across a wide range of tasks, such as classification [19, 37], clustering [31, 40], retrieval [5, 43], and recognition [35, 44].

Although effective, existing GNNs typically require a prior graph to learn node representations, which poses a major challenge when encountering incomplete or even missing graphs. This limitation has spurred the development of latent graph inference (LGI) [7, 10, 17, 22, 32], also known as graph structure learning [9, 24, 39, 42]. LGI aims to jointly learn the underlying graph and discriminative node representations solely from the features of nodes in an end-to-end fashion. By adaptively learning the graph topology, LGI models are empowered with great ability to remove noise and capture more complex structure of the data [13, 26, 27, 47]. Consequently, LGI emerges as a promising research topic with a broad range of applications, such as point cloud segmentation [41], disease prediction [6], multi-view clustering [31], and brain connectome representation [18].

---

*Corresponding author: JianglinLu@outlook.com.

37th Conference on Neural Information Processing Systems (NeurIPS 2023).

However, many LGI methods suffer from a so-called *supervision starvation* (SS) problem [10], where a number of edge weights are learned without any semantic supervision during the graph inference stage. Specifically, given a $k$-layer GNN, if a node and all of its predefined neighbors (from 1- to $k$-hop) are unlabeled, the edge weights between this node and its neighbors will not contribute to the training loss and cannot be semantically optimal after training. This will lead to poor generalization performance since these under-trained weights are inevitably used to make predictions for testing samples. In this paper, we discover that the SS problem is actually caused by the graph sparsification operation, which severely destroys the important connections established between pivotal nodes and labeled ones. Based on this observation, we propose to *restore the destroyed connections and replenish the missed supervision for better LGI*.

Specifically, we first define the pivotal nodes as *$k$-hop starved nodes*. Then, the SS problem can be transformed into a more manageable task of eliminating starved nodes. We propose to identify the $k$-hop starved nodes based on the $k$-th power of a given adjacency matrix. After identification, we can diminish the starved nodes by incorporating a regularization adjacency matrix into the initial one. However, the above identification approach suffers from high computational complexity due to matrix multiplication. For example, given a 2-layer GNN, identifying 2-hop starved nodes requires at least $\mathcal{O}(n^3)$, where $n$ denotes the number of nodes. To address this, we present a more efficient alternative solution inspired by *CUR matrix decomposition* [2, 3]. We find that with appropriate column and row selection, the $U$ matrix obtained through CUR decomposition of the initial adjacency matrix is actually a zero matrix. Thus, we can eliminate the starved nodes by reconstructing the $U$ matrix.

Although recovering the $U$ matrix encourages more supervision, such strategy may cause a potential issue we call *weight contribution rate decay*. In other words, the weight contribution rate (WCR) of non-starved nodes decays as the number of starved nodes increases, resulting in a final latent graph that heavily relies on the regularization one. To tackle this issue, we propose two simple strategies, *i.e.*, decreasing the WCR of starved nodes and increasing the WCR of non-starved nodes (see Sec. 3.3 for details). The main contributions of this paper can be summarized as follows:

- We identify that graph sparsification is the main cause of the supervision starvation (SS) problem in latent graph inference (LGI). Therefore, we propose to restore the affinities corrupted by the sparsification operation to replenish the missed supervision for better LGI.

- By defining $k$-hop starved nodes, we transform the SS issue into a more manageable task of removing starved nodes. Inspired by CUR decomposition, we propose a simple yet effective solution to determine the starved nodes and diminish them using a regularization graph.

- The proposed approach is model-agnostic and can be seamlessly integrated into various LGI models. Extensive experiments demonstrate that eliminating the starved nodes consistently enhances the state-of-the-art LGI methods, particularly under extremely limited supervision.

## 2 Preliminaries

### 2.1 Notations

Let $\mathbf{A}_{i:}$, $\mathbf{A}_{:j}$, $\mathbf{A}_{ij}$, and $\mathbf{A}^k$ denote the $i$-th row, the $j$-th column, the element at the $i$-th row and $j$-th column, and the $k$-th power of the matrix $\mathbf{A}$, respectively. Let $\mathbf{1}_n$ represent an $n$-dimensional column vector with all elements being 1. Let $\mathbb{1}_{\mathbb{R}+}(\mathbf{A})$ be an element-wise indicator function that sets $\mathbf{A}_{ij}$ to 1 if it is positive, and 0 otherwise. Conversely, $\mathbb{1}_{\mathbb{R}-}(\mathbf{A})$ sets $\mathbf{A}_{ij}$ to 0 if it is positive, and 1 otherwise.

### 2.2 Latent Graph Inference

**Definition 1** (Latent Graph Inference). *Given a graph $\mathcal{G}(\mathcal{V}, \mathbf{X})$ containing $n$ nodes $\mathcal{V} = \{V_1, \ldots, V_n\}$ and a feature matrix $\mathbf{X} \in \mathbb{R}^{n \times d}$ with each row $\mathbf{X}_{i:} \in \mathbb{R}^d$ representing the $d$-dimensional attributes of node $V_i$, latent graph inference (LGI) aims to simultaneously learn the underlying graph topology encoded by an adjacency matrix $\mathbf{A} \in \mathbb{R}^{n \times n}$ and the discriminative $d'$-dimensional node representations $\mathbf{Z} \in \mathbb{R}^{n \times d'}$ based on $\mathbf{X}$, where the learned $\mathbf{A}$ and $\mathbf{Z}$ are jointly optimal for certain downstream tasks $\mathcal{T}$ given a specific loss function $\mathcal{L}$.*

In general, an LGI model mainly consists of a latent graph generator $\mathcal{P}_\Phi(\mathbf{X})$: $\mathbb{R}^{n \times d} \to \mathbb{R}^{n \times n}$ that generates an adjacency matrix $\mathbf{A}$ based on the node features $\mathbf{X}$, and a node encoder $\mathcal{F}_\Theta(\mathbf{X}, \mathbf{A})$:

$\mathbb{R}^{n \times d} \to \mathbb{R}^{n \times d'}$ that learns discriminative node representations $\mathbf{Z}$ based on $\mathbf{X}$ and the learned $\mathbf{A}$. In practice, $\mathcal{F}_{\Theta}$ is typically implemented using a $k$-layer GNN, while $\mathcal{P}_{\Phi}$ can be realized through different strategies such as full parameterization [17, 36], MLP [10, 25], or attentive network [4, 15]. In this paper, we adopt the most common settings from existing LGI literature [4, 10, 13, 25, 46], considering $\mathcal{T}$ as the semi-supervised node classification task and $\mathcal{L}$ as the cross-entropy loss.

## 2.3 Supervision Starvation

To illustrate the supervision starvation problem [10], we consider a general LGI model $\mathcal{M}$ consisting of a latent graph generator $\mathcal{P}_{\Phi}$ and a node encoder $\mathcal{F}_{\Theta}$. For simplicity, we ignore the activation function and assume that $\mathcal{F}_{\Theta}$ is implemented using a 1-layer GNN, *i.e.*, $\mathcal{F}_{\Theta} = \text{GNN}_1(\mathbf{X}, \mathbf{A}; \Theta)$, where $\mathbf{A} = \mathcal{P}_{\Phi}(\mathbf{X})$. For each node $\mathbf{X}_{i:}$, the corresponding node representation $\mathbf{Z}_{i:}$ learned by the model $\mathcal{M}$ can be expressed as:

$$\mathbf{Z}_{i:} = \mathbf{A}_{i:}\mathbf{X}\Theta = \left( \sum_{j \in \Omega} \mathbf{A}_{ij}\mathbf{X}_{j:} \right) \Theta, \tag{1}$$

where $\Omega = \{j \mid \mathbb{1}_{\mathbb{R}^+}(\mathbf{A})_{ij} = 1\}$ and $\mathbf{A}_{ij} = \mathcal{P}_{\Phi}(\mathbf{X}_{i:}, \mathbf{X}_{j:})$. Consider the node classification loss:

$$\min_{\mathbf{A}, \Theta} \mathcal{L} = \sum_{i \in \mathcal{Y}_L} \sum_{j=1}^{|\mathcal{C}|} \mathbf{Y}_{ij} \ln \mathbf{Z}_{ij} = \sum_{i \in \mathcal{Y}_L} \mathbf{Y}_{i:} \ln \mathbf{Z}_{i:}^{\top} = \sum_{i \in \mathcal{Y}_L} \mathbf{Y}_{i:} \ln \left( \left( \sum_{j \in \Omega} \mathbf{A}_{ij}\mathbf{X}_{j:} \right) \Theta \right)^{\top}, \tag{2}$$

where $\mathcal{Y}_L$ represents the set of indexes of labeled nodes and $|\mathcal{C}|$ denotes the size of label set. For $\forall i \in \mathcal{Y}_L$, $j \in \Omega$, $\mathbf{A}_{ij}$ is optimized via backpropagation under the supervision of label $\mathbf{Y}_{i:}$. For $\forall i \notin \mathcal{Y}_L$, however, if $j \notin \mathcal{Y}_L$ for $\forall j \in \Omega$, $\mathbf{A}_{ij}$ will receive no supervision from any label and, as a result, cannot be semantically optimal after training. Consequently, the learning models exhibit poor generalization as the predictions of testing nodes inevitably rely on these supervision-starved weights. This phenomenon is referred to as *supervision starvation* (SS), where many edge weights are learned without any label supervision. It is easy to infer that this issue also persists in a $k$-layer GNN.

*We may ask why this problem arises?* In fact, the SS problem is caused by a common and necessary post-processing operation known as graph sparsification, which is employed in the majority of LGI methods [10, 25, 38, 46] to generate a sparse latent graph. To be more specific, graph sparsification adjusts the initial dense graph to a sparse one through the following procedure:

$$\mathbf{A}_{ij} = \begin{cases} \mathbf{A}_{ij}, & \text{if } \mathbf{A}_{ij} \in \text{top-}\kappa(\mathbf{A}_{i:}) \\ 0, & \text{otherwise}, \end{cases} \tag{3}$$

where $\text{top-}\kappa(\mathbf{A}_{i:})$ denotes the set of the top $\kappa$ values in $\mathbf{A}_{i:}$. After this sparsification operation, a significant number of edge weights are directly erased, including the crucial connections established between pivotal nodes and labeled nodes. *Another question that may arise is: how many important nodes or connections suffer from this problem?* We delve into this question in the next section.

## 3 Methodology

### 3.1 Identification & Elimination of Starved Nodes

We first introduce the definitions of $k$-hop starved node and the corresponding $k$-hop starved weight.

**Definition 2** ($k$-hop Starved Node). *Given a graph $\mathcal{G}(\mathcal{V}, \mathbf{X})$ consisting of $n$ nodes $\mathcal{V} = \{V_1, \ldots, V_n\}$ and the corresponding node features $\mathbf{X}$, for a $k$-layer graph neural network $\text{GNN}_k(\mathbf{X}; \Theta)$ with network parameters $\Theta$, the unlabeled node $V_i$ is a $k$-hop starved node if, for $\forall \kappa \in \{1, \ldots, k\}$, $\forall V_j \in \mathcal{N}_{\kappa}(i)$, where $\mathcal{N}_{\kappa}(i)$ is the set of $\kappa$-hop neighbors of $V_i$, $V_j$ is unlabeled. Specifically, 0-hop starved nodes are defined as the unlabeled nodes*[2].

Based on Definition 2, we can define the $k$-hop starved weight as follows.

---

[2]Here, we want to clarify that self-connections are not considered when defining $k$-hop neighbors.

**Definition 3** (*$k$-hop Starved Weight*). *If an edge exists between nodes $V_i$ and $V_j$, the associated edge weight $\mathbf{A}_{ij}$ is a $k$-hop starved weight if both of the $V_i$ and $V_j$ qualifies as $(k-1)$-hop starved nodes*[3].

According to the definition presented, it is evident that $k$-hop starved weights are precisely the ones that receive no semantic supervision from labels. As a result, to address the SS problem, our focus should be on reducing the presence of $k$-hop starved nodes. The following theorem illustrates how we can identify such nodes based on a given initial adjacency matrix.

**Theorem 1.** *Given a sparse adjacency matrix $\mathbf{A} \in \mathbb{R}^{n \times n}$ with self-connections generated on graph $\mathcal{G}(\mathcal{V}, \mathbf{X})$ by a latent graph inference model with a $k$-layer graph neural network $\mathtt{GNN}_k(\mathbf{X}; \mathbf{\Theta})$, the node $V_i$ is a $k$-hop starved node, if $\exists j \in \{1, \ldots, n\}$, such that $[\mathbb{1}_{\mathbb{R}+}(\mathbf{A})]_{ij}^k = 1$, and for $\forall j \in \{j \mid [\mathbb{1}_{\mathbb{R}+}(\mathbf{A})]_{ij} = 1 \cup [\mathbb{1}_{\mathbb{R}+}(\mathbf{A})]_{ij}^2 = 1 \cup \ldots \cup [\mathbb{1}_{\mathbb{R}+}(\mathbf{A})]_{ij}^k = 1\}$, $V_j$ is unlabeled.*

*Proof.* Please refer to the supplementary material for details. ☐

To provide an intuitive perspective, we use two real-world graph datasets, namely Cora (2708 nodes) and Citeseer (3327 nodes), as examples. We calculate the number of $k$-hop starved nodes for $k = 1, 2, 3, 4$, based on their original graph topology. Fig. 1 shows the statistical results for the Cora140, Cora390, Citesser120, and Citeseer370 datasets, where the suffix number represents the number of labeled nodes. From Fig. 1, we observe that as the value of $k$ increases, the number of starved nodes decreases. This can be explained by the fact that as $k$ increases, the nodes have more neighbors (from 1- to $k$-hop), and the possibility of having at least one labeled neighbor increases. Adopting a deeper GNN (larger $k$) can thus mitigate the SS problem. However, it is important to consider that deeper GNNs result in higher computational consumption and may lead to poorer general-

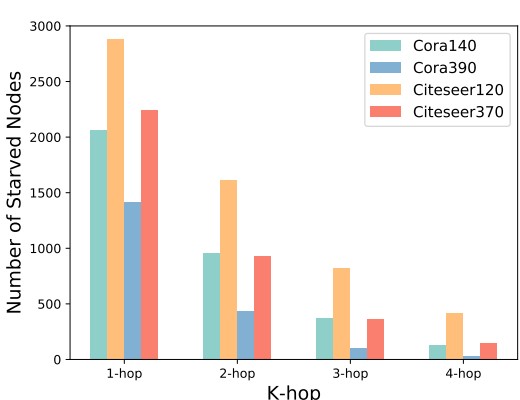

Figure 1: Illustration of $k$-hop starved nodes on different datasets. Obviously, the number of $k$-hop starved nodes decreases as the value of $k$ increases.

ization performance [1, 21, 30]. Furthermore, as shown in Fig. 1, even with a 4-layer GNN, there are still hundreds of 4-hop starved nodes in the Citesser120. Therefore, we believe that employing a deeper GNN is not the optimal solution to resolve the SS problem [4].

In fact, Theorem 1 implicitly indicates how to alleviate the SS problem. Intuitively, a straightforward solution is to ameliorate the given adjacency matrix $\mathbf{A}$ in order to reduce the number of starved nodes. This can be accomplished simply by reconstructing the connections between starved nodes and the labeled ones. Technically, we can achieve this by adding a regularization adjacency matrix $\mathbf{B}$ to $\mathbf{A}$:

$$\widetilde{\mathbf{A}} = \mathbf{A} + \alpha\mathbf{B}, \tag{4}$$

where $\widetilde{\mathbf{A}}$ is the refined adjacency matrix, $\mathbf{B}$ models the recovered affinities between staved nodes and labeled ones, and $\alpha$ is a balanced parameter that controls the contribution of $\mathbf{A}$ and $\mathbf{B}$. According to Theorem 1, we can identify the starved nodes based on $\mathbf{A}$ and replenish the missed supervision through $\mathbf{B}$, thereby preventing $\widetilde{\mathbf{A}}$ from being starved. Specifically, for each starved node $V_i$, we can search for at least one closest labeled node $V_l$, and restore at least one connection between $V_l$ and $V_j$ for $j \in \{\mathcal{N}_\kappa(i) \cup i\}$ such that $[\mathbb{1}_{\mathbb{R}+}(\mathbf{B})]_{jl}^\kappa = 1$, where $\kappa$ can be arbitrarily chosen from the set of $\{1, 2, \ldots, k\}$. Although this strategy is effective, it may be computationally complex. Even with a small value of $k$, the computational cost of identifying $k$-hop starved nodes based on Theorem 1 is prohibitively expensive. For example, when identifying 2-hop starved nodes, the time complexity of computing $\mathbf{A}^2$ alone reaches $\mathcal{O}(n^3)$. In the next section, we will propose a more efficient solution.

---

[3]Note that, for a $k$-layer GNN, a $k$-hop starved node also qualifies as a $(k-1)$-hop starved node.
[4]We discuss this further in the supplementary material.

## 3.2 CUR Decomposition Makes Better Solution

Inspired by CUR matrix decomposition [2, 3], we propose an efficient alternative approach to identify the starved nodes. We first present the definition of CUR matrix decomposition.

**Definition 4** (CUR Decomposition [2]). *Given $\mathbf{Q} \in \mathbb{R}^{n \times m}$ of rank $\rho = \mathtt{rank}(\mathbf{Q})$, rank parameter $k < \rho$, and accuracy parameter $0 < \varepsilon < 1$, construct column matrix $\mathbf{C} \in \mathbb{R}^{n \times c}$ with $c$ columns from $\mathbf{Q}$, row matrix $\mathbf{R} \in \mathbb{R}^{r \times m}$ with $r$ rows from $\mathbf{Q}$, and intersection matrix $\mathbf{U} \in \mathbb{R}^{c \times r}$ with $c$, $r$, and $\mathtt{rank}(\mathbf{U})$ being as small as possible, in oder to reconstruct $\mathbf{Q}$ within relative-error:*

$$||\mathbf{Q} - \mathbf{CUR}||_F^2 \leq (1 + \varepsilon)||\mathbf{Q} - \mathbf{Q}_k||_F^2. \tag{5}$$

*Here, $\mathbf{Q}_k = \mathbf{U}_k \mathbf{\Sigma}_k \mathbf{V}_k^T \in \mathbb{R}^{n \times m}$ is the best rank $k$ matrix obtained via the singular value decomposition (SVD) of $\mathbf{Q}$.*

With the definition of CUR decomposition, we can find a more efficient solution to identify the starved nodes. The following theorem demonstrates how we can accomplish this goal.

**Theorem 2.** *Given a sparse adjacency matrix $\mathbf{A} \in \mathbb{R}^{n \times n}$ with self-connections generated on graph $\mathcal{G}(\mathcal{V}, \mathbf{X})$, construct $\mathbf{C} = \mathbf{A}[:, col\_mask] \in \mathbb{R}^{n \times c}$, where $col\_mask \in \{0, 1\}^n$ contains only $c$ positive values corresponding to $c$ labeled nodes, and $\mathbf{R} = \mathbf{A}[row\_mask, :] \in \mathbb{R}^{r \times n}$ with $row\_mask = \mathbb{1}_{\mathbb{R}-}(\mathbf{C}\mathbb{1}_c) \in \{0, 1\}^n$. Then, (a) $\mathbf{U} = \mathbf{A}[row\_mask, col\_mask] = \mathbf{0} \in \mathbb{R}^{r \times c}$, where $\mathbf{0}$ is a zero matrix, (b) the set of 1-hop starved nodes $\mathtt{Set}_1(r) = \{V_i | i \in RM_+\}$, where $RM_+ \in \mathbb{N}^r$ indicates the set of indexes of positive elements from $row\_mask$, and (c) for each $i \in RM_+$, $V_i$ is a 2-hop starved node if, for $\forall j$ satisfying $[\mathbb{1}_{\mathbb{R}+}(\mathbf{R})]_{ij} = 1$, $j \in RM_+$.*

*Proof.* Please refer to the supplementary material for details. $\square$

Theorem 2 provides a more efficient alternative for identifying $k$-hop starved nodes for $k \in \{1, 2\}$. In fact, the column matrix $\mathbf{C}$ models the relationships between all nodes and $c$ labeled nodes, the row matrix $\mathbf{R}$ models the affinities between $r$ 1-hop starved nodes and the whole nodes, and the intersection matrix $\mathbf{U}$ models the strength of connections between $r$ 1-hop starved nodes and $c$ labeled nodes. Theorem 2 states that $\mathbf{U} = \mathbf{0}$, indicting that there are no connections between the starved nodes and the labeled ones. Based on this observation, we propose a simpler approach to reduce the number of starved nodes. Specifically, we rebuild the intersection matrix $\mathbf{U}$ to ensure that the reconstructed $\widetilde{\mathbf{U}} \neq \mathbf{0}^5$. Consequently, Eq. (4) can be rewritten as:

$$\widetilde{\mathbf{A}} = \mathbf{A} + \alpha \mathbf{B} = \mathbf{A} + \alpha \Gamma \left( \widetilde{\mathbf{U}}, n \right), \tag{6}$$

where function $\Gamma(\widetilde{\mathbf{U}}, n)$ extends the matrix $\widetilde{\mathbf{U}} \in \mathbb{R}^{r \times c}$ to an $n \times n$ matrix by padding $n - r$ rows of zeros and $n - c$ columns of zeros in the corresponding positions.

The question now turns to how to reconstruct the intersection matrix $\widetilde{\mathbf{U}}$. For simplicity, we directly adopt the same strategy used in constructing $\mathbf{A}$. Specifically, for each row $i$ of $\widetilde{\mathbf{U}}$, we establish a connection between $V_i$ and $V_j$ for $\forall j \in CM_+$, where $CM_+ \in \mathbb{N}^c$ represents the set of indexes of positive elements from $col\_mask$. We then assign weights to these connections based on their distance. Note that, if we ensure each row of $\widetilde{\mathbf{U}}$ has at least one weight greater than 0, there will be no $\kappa$-hop starved nodes for $\forall \kappa > 1$. This means that we do not need to feed the $k$-hop starved nodes satiated, simply feeding $\kappa$-hop ones for $\forall \kappa < k$ makes $k$-hop starved nodes cease to exist. In addition, compared with the time complexity of $\mathcal{O}(dn^2)$ to construct $\mathbf{A}$, the time complexity of reconstructing $\widetilde{\mathbf{U}}$ is $\mathcal{O}(drc)$. The additional computational burden is relatively negligible since $c \ll n$.

## 3.3 Weight Contribution Rate Decay

Directly replenishing the additional supervision encoded in $\widetilde{\mathbf{U}}$ by Eq. (6) may cause a potential issue we refer to as *weight contribution rate decay*. To understand this issue, we consider the $j$-th column

---

[5]In fact, with the reconstructed $\widetilde{\mathbf{U}}$, we can set $\widetilde{\mathbf{Q}} = \mathbf{C}\widetilde{\mathbf{U}}\mathbf{R}$ as the regularization $\mathbf{B}$. It is, of course, sensible and feasible. However, a potential drawback is that the reconstruction of $\widetilde{\mathbf{Q}}$ requires matrix multiplications of three matrices, which is time-consuming. Unexpectedly, we find that only constructing matrix $\widetilde{\mathbf{U}}$ is enough to solve the SS problem since it models the relationships between 1-hop starved nodes and labeled ones.

vector $\widetilde{\mathbf{A}}_{:j}$ of $\widetilde{\mathbf{A}}$ for $j \in \mathtt{CM}_+$. We define the weight contribution rates (WCR) of non-starved nodes $\rho_-$ and the WCR of starved nodes $\rho_+$ as:

$$\rho_- = \frac{\sum_{i \notin \mathtt{RM}_+} \widetilde{\mathbf{A}}_{ij}}{\sum_i \widetilde{\mathbf{A}}_{ij}} = \frac{\sum_{i \notin \mathtt{RM}_+} \mathbf{A}_{ij}}{\sum_i \widetilde{\mathbf{A}}_{ij}}, \quad \rho_+ = 1 - \rho_- = 1 - \frac{\sum_{i \notin \mathtt{RM}_+} \mathbf{A}_{ij}}{\sum_i \widetilde{\mathbf{A}}_{ij}} = \frac{\sum_{i \in \mathtt{RM}_+} \alpha \widetilde{\mathbf{U}}_{ij}}{\sum_i \widetilde{\mathbf{A}}_{ij}}. \quad (7)$$

From Eq. (7), we observe that the WCR of non-starved nodes $\rho_-$ decays as the number of starved nodes increases (*i.e.*, $\rho_+$ increases). This means that $\rho_-$ becomes negligible if there are numerous starved nodes. As a result, the $j$-th column vector $\widetilde{\mathbf{A}}_{:j}$ of $\widetilde{\mathbf{A}}$ for $j \in \mathtt{CM}_+$ heavily relies on the $j$-th column vector $\widetilde{\mathbf{U}}_{:j}$ of the reconstructed intersection matrix $\widetilde{\mathbf{U}}$. This outcome is not desirable since the regularization imposed should not dominate a significant portion of the final adjacency matrix. Recalling our initial objective of properly refining the original latent graph to replenish the missed supervision, we will design two simple strategies to relieve this issue.

**Decrease $\rho_+$.** On the one hand, we can decrease the WCR of starved nodes $\rho_+$ by selecting only $\tau(\tau < c)$ labeled nodes as the supplementary adjacent points for each 1-hop starved node. This results in a sparse intersection matrix $\widehat{\mathbf{U}}$. For this strategy, we present the following proposition:

**Proposition 1.** *Suppose that we randomly select $\tau$ out of $c$ labeled nodes as the supplementary adjacent points for each 1-hop starved node. If there are $r$ 1-hop starved nodes, then for $\forall j \in \mathtt{CM}_+$, we have $\widehat{\rho}_+ = \frac{\sum_{i \in \mathtt{RM}_+} \alpha \widehat{\mathbf{U}}_{ij}}{\sum_i \widetilde{\mathbf{A}}_{ij}} \leq \rho_+$, where $\widehat{\rho}_+ = \rho_+$ with only a probability of $\left(\frac{\tau}{c}\right)^r$.*

*Proof.* Please refer to the supplementary material for details. $\square$

**Increase $\rho_-$.** On the other hand, we can improve the WCR of non-starved nodes $\rho_-$ by magnifying the weights of non-starved nodes. Specifically, we construct an additional regularization matrix $\mathbf{Q} \in \mathbb{R}^{n \times c}$, where its $c$ columns correspond to the $c$ labeled nodes and $\mathbf{Q}_{ij} = 0$ for $\forall i \in \mathtt{RM}_+$. For $\forall i \notin \mathtt{RM}_+$, we establish connections between the node $V_i$ and $\tau(\tau < c)$ labeled nodes $V_j$, and assign the corresponding weights $\mathbf{Q}_{ij}$ using a similar strategy as in constructing $\widehat{\mathbf{U}}$. By adding the additional regularization matrix $\mathbf{Q}$, the refined adjacency matrix $\widehat{\mathbf{A}}$ can be expressed as:

$$\widehat{\mathbf{A}} = \mathbf{A} + \alpha \left( \Gamma \left( \widehat{\mathbf{U}}, n \right) + \Gamma \left( \mathbf{Q}, n \right) \right). \quad (8)$$

Similarly, for this strategy, we have the following proposition:

**Proposition 2.** *Suppose that we randomly select $\tau$ out of $c$ labeled nodes as the supplementary adjacent points for each non-starved node. If there are $r$ 1-hop starved nodes, then for $\forall j \in \mathtt{CM}_+$, we have $\widehat{\rho}_- = \frac{\sum_{i \notin \mathtt{RM}_+} \widehat{\mathbf{A}}_{ij}}{\sum_i \widehat{\mathbf{A}}_{ij}} = \frac{\sum_{i \notin \mathtt{RM}_+} \widetilde{\mathbf{A}}_{ij} + \alpha \sum_{i \notin \mathtt{RM}_+} \mathbf{Q}_{ij}}{\sum_i \widetilde{\mathbf{A}}_{ij} + \alpha \sum_{i \notin \mathtt{RM}_+} \mathbf{Q}_{ij}} \geq \rho_-$, where $\widehat{\rho}_- = \rho_-$ with only a probability of $\left(1 - \frac{\tau}{c}\right)^{n-r}$.*

*Proof.* Please refer to the supplementary material for details. $\square$

### 3.4 End-to-End Training

Note that the proposed approach is model-agnostic and can be seamlessly integrated into existing LGI models. In Sec. 4, we will apply our design to state-of-the-art LGI methods [4, 10, 13, 46] and compare its performance with the original approach. Therefore, we follow these methods and implement the node encoder $\mathcal{F}_\Theta$ using a 2-layer GNN. Then, we calculate the cross-entropy loss $\mathcal{L}_{\mathrm{ce}}$ between the true labels $\mathbf{Y}$ and the predictions $\mathbf{Z}$, as well as a graph regularization loss $\mathcal{L}_{\mathrm{reg}}$ on $\widehat{\mathbf{A}}$:

$$\min \mathcal{L} = \mathcal{L}_{\mathrm{ce}} + \gamma \mathcal{L}_{\mathrm{reg}} = \sum_{i \in \mathcal{Y}_L} \sum_{j=1}^{|\mathcal{C}|} \mathbf{Y}_{ij} \ln \mathbf{Z}_{ij} + \gamma \mathcal{L}_{\mathrm{reg}}, \quad (9)$$

where $\gamma$ is a balanced parameter and $\mathbf{Z} = \mathcal{F}_\Theta(\mathbf{X}, \widehat{\mathbf{A}}) = \mathrm{softmax}\left( \widehat{\mathbf{A}} \sigma \left( \widehat{\mathbf{A}} \mathbf{X} \Theta^0 \right) \Theta^1 \right)$. According to different LGI methods, the graph regularization loss $\mathcal{L}_{\mathrm{reg}}$ can be different, such as Dirichlet energy [4] and self-supervision [10] (see supplementary material for more details). For fairness, in the experiments, we will adopt the same graph regularization loss as the comparison methods.

Table 1: Test accuracy (%) of the baselines (M) and our CUR extension versions (M_U and M_R) on various datasets with different labeling rates (marked in **bold**), where "OOM" indicates out of memory. The highest and second highest results are marked in red and blue, respectively.

| Models / datasets | ogbn-arxiv | Cora390 | Cora140 | Citeseer370 | Citeseer120 | Pubmed |
|---|---|---|---|---|---|---|
| # of labeled / all nodes | 90941/169343 | 390/2708 | 140/2708 | 370/3327 | 120/3327 | 60/19717 |
| **Labeling rate** | **53.70%** | **14.40%** | **5.17%** | **11.12%** | **3.61%** | **0.30%** |
| GCN+KNN | 55.15 ± 0.11 | 72.82 ± 0.39 | 67.94 ± 0.29 | 73.28 ± 0.23 | 69.68 ± 0.53 | 68.66 ± 0.05 |
| GCN+KNN_U (ours) | 55.82 ± 0.11 | 72.82 ± 0.21 | 68.18 ± 0.44 | 73.68 ± 0.10 | 69.74 ± 0.54 | 74.12 ± 0.32 |
| GCN+KNN_R (ours) | 55.86 ± 0.10 | 72.92 ± 0.28 | 68.12 ± 0.48 | 73.66 ± 0.14 | 69.90 ± 0.68 | 74.78 ± 0.17 |
| GCN&KNN | OOM | 72.16 ± 0.54 | 68.76 ± 1.20 | 77.28 ± 0.64 | 68.64 ± 1.14 | OOM |
| GCN&KNN_U (ours) | OOM | 73.04 ± 0.20 | 70.16 ± 0.91 | 78.40 ± 0.44 | 70.52 ± 1.04 | OOM |
| GCN&KNN_R (ours) | OOM | 73.20 ± 0.25 | 70.24 ± 0.97 | 78.48 ± 0.30 | 69.48 ± 0.77 | OOM |
| IDGL [4] | OOM | 74.00 ± 0.38 | 70.74 ± 0.50 | 71.30 ± 0.17 | 69.24 ± 0.19 | OOM |
| IDGL_U (ours) | OOM | 74.54 ± 0.52 | 70.82 ± 0.49 | 72.46 ± 0.14 | 69.32 ± 0.39 | OOM |
| IDGL_R (ours) | OOM | 74.48 ± 0.47 | 71.14 ± 0.22 | 72.56 ± 0.12 | 69.86 ± 0.50 | OOM |
| LCGS [13] | OOM | 72.02 ± 0.37 | 69.88 ± 0.66 | 73.84 ± 0.83 | 72.30 ± 0.33 | OOM |
| LCGS_U (ours) | OOM | 72.18 ± 0.31 | 70.04 ± 0.80 | 74.18 ± 0.43 | 72.38 ± 0.43 | OOM |
| LCGS_R (ours) | OOM | 72.22 ± 0.45 | 70.14 ± 0.64 | 74.20 ± 0.36 | 72.40 ± 0.42 | OOM |
| GRCN [46] | OOM | 73.34 ± 0.27 | 68.86 ± 0.25 | 73.62 ± 0.23 | 71.24 ± 0.19 | 69.24 ± 0.20 |
| GRCN_U (ours) | OOM | 74.10 ± 0.25 | 69.44 ± 0.34 | 73.88 ± 0.34 | 71.54 ± 0.31 | 72.80 ± 0.99 |
| GRCN_R (ours) | OOM | 74.14 ± 0.22 | 69.56 ± 0.22 | 74.22 ± 0.13 | 71.64 ± 0.41 | 72.82 ± 1.03 |
| SLAPS [10] | 55.46 ± 0.12 | 76.62 ± 0.83 | 74.26 ± 0.53 | 74.32 ± 0.56 | 70.66 ± 0.97 | 74.86 ± 0.79 |
| SLAPS_U (ours) | 55.68 ± 0.09 | 76.94 ± 0.42 | 74.56 ± 0.21 | 74.82 ± 0.27 | 71.68 ± 0.47 | 76.74 ± 0.59 |
| SLAPS_R (ours) | 56.11 ± 0.15 | 76.82 ± 0.19 | 75.00 ± 0.49 | 74.90 ± 0.42 | 72.36 ± 0.49 | 77.12 ± 0.77 |

## 4 Experiments

### 4.1 Experimental Settings

**Baselines.** As mentioned earlier, the proposed regularization module can be easily integrated into most existing LGI methods. To evaluate its effectiveness, we select representative LGI methods as baselines, including IDGL [4], GRCN [46], SLAPS [10], and LCGS [13]. We also consider two additional baselines marked as GCN+KNN [10, 19] and GCN&KNN [4, 19]. GCN+KNN is a two-step method that first constructs a KNN graph based on feature similarities and then feeds the pre-constructed graph to a GCN for training. GCN&KNN is an end-to-end method that learns the latent graph and network parameters simultaneously. For methods that require a prior graph, we use the same KNN graph as in GCN+KNN. For GCN&KNN, we simply adopt the Dirichlet energy [4] as the graph regularization loss.

**Datasets.** Following the common settings of existing LGI methods [4, 10, 13, 46], we conduct experiments on four well-known benchmarks: Cora, Citeseer, Pubmed [19, 34], and ogbn-arxiv [14]. For detailed dataset statistics, please refer to the supplementary material. For all datasets, we only provide the original node features for training. To test the performance under different labeling rates, for the Cora and Citeseer datasets, we add half of the validation samples to the training sets, resulting in Cora390 and Citeseer370, where the suffix number represents the total number of labeled nodes.

**Implementation.** We compare the above baselines with their corresponding CUR extension versions. Specifically, we consider two CUR extensions termed M_U and M_R (M refers to the baseline), where the former adopts the sparse intersection matrix $\widehat{\mathbf{U}}$ as the regularization, and the latter combines $\widehat{\mathbf{U}}$ and $\mathbf{Q}$ together (see Sec. 3.3 for details). In experiments, we practically select the $\tau$ closest labeled nodes as supplementary adjacent points for each row of $\widehat{\mathbf{U}}$ and $\mathbf{Q}$. For postprocessing operations on the graph, such as symmetrization and normalization [4, 10], we follow the baselines and adopt the same operations for fairness. We select the values of $\tau$ and $\alpha$ from the sets $\{10, 15, 20, 25, 30, 50\}$ and $\{0.01, 0.1, 1.0, 10, 50, 100\}$, respectively. For other hyperparameters such as learning rate and weight decay, we follow the baselines and use the same settings. For each method, we record the best testing performance and report the average accuracy of five independent experiments, along with the corresponding standard deviation.

### 4.2 Comparison Results

Table 1 presents the comparison results on all used datasets. It is evident that our proposed CUR extensions consistently outperform the corresponding baselines, demonstrating the effectiveness of eliminating starved nodes. When considering the labeling rates of different datasets listed in the third row of Table 1, we observe that the lower the labeling rate, the greater the performance

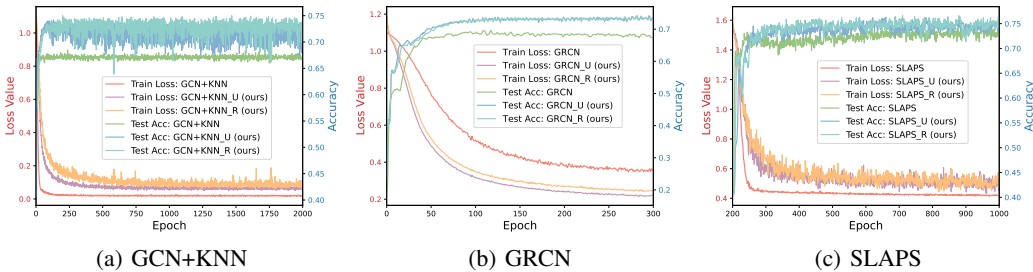

|            | (a) GCN+KNN | (b) GRCN | (c) SLAPS |
|---|---|---|---|

Figure 2: Training loss (left vertical axis) and testing accuracy (right vertical axis) curves of GCN+KNN, GRCN, SLAPS, and their corresponding CUR extensions on the Pubmed dataset.

Table 2: Number of $k$-hop starved nodes ($k \in \{1, 2\}$) on various datasets when selecting different number of neighbors ($\tau$) in graph sparsification operation.

| Number of neighbors ($\tau$) | | 10 | | | 20 | |
|---|---|---|---|---|---|---|
| Datasets | Cora140 | Citeseer120 | Pubmed | Cora140 | Citeseer120 | Pubmed |
| 1-hop starved nodes | 1,625 | 2,268 | 19,076 | 993 | 1,577 | 18,478 |
| 2-hop starved nodes | 100 | 299 | 16,756 | 0 | 0 | 12,843 |

improvement achieved by our methods. Notably, on the Pubmed dataset with an extremely low labeling rate of $0.30\%$, the accuracy of our proposed methods (M_R) increase by $6.12\%$, $3.58\%$, and $2.26\%$ compared to the basic models (M) of GCN+KNN, GRCN, and SLAPS, respectively. This is because the lower the labeling rate, the more starved nodes exist in the dataset (see Fig. 1 for an example). Our proposed methods aim to restore the destroyed affinities between starved nodes and labeled nodes, enabling us to leverage the semantic supervision missed by baselines, thereby achieving superior performance with extremely limited supervision.

Fig. 2 displays the training loss and testing accuracy curves of GCN+KNN, GRCN, SLAPS, and their CUR extensions on the Pubmed dataset. We omit the curves for the first 200 epochs of SLAPS because, during this stage, SLAPS focuses solely on learning the latent graph through an additional self-supervision task without involving the classification task. From Fig. 2, we observe that our proposed CUR extensions achieve higher testing accuracy compared to their corresponding baselines. In comparison with the CUR extensions of GCN+KNN, the CUR extensions of GRCN and SLAPS exhibit relatively stable accuracy results as the training epoch increases. This stability can be attributed to GRCN and SLAPS jointly learning the latent graph and network parameters in an end-to-end manner, whereas GCN+KNN pre-constructs a KNN graph and maintains a fixed graph during training without optimization. For GRCN, our proposed CUR extensions demonstrate lower training loss and higher testing accuracy, further indicating their improved generalization by removing starved nodes. Additionally, we observe relatively unstable training loss for the CUR extensions of SLAPS. The potential reason is that SLAPS introduces an additional graph regularization loss through a self-supervision task, while its CUR extensions aim to reconstruct the destroyed connections in the latent graph. As a result, the training process for the self-supervision task can exhibit some instability.

## 4.3 Discussion

We would like to explore the question that *why our proposed methods yield a slight improvement on the Cora and Citeseer datasets, while achieving a substantial improvement on the Pubmed dataset*. As shown in Table 2, when $\tau = 10$, there are 100 and 299 2-hop starved nodes on the Cora140 and Citeseer120 datasets, respectively. However, when $\tau = 20$, there are no 2-hop starved nodes on the Cora140 and Citeseer120 datasets, and the number of 1-hop starved nodes also sharply decreases. In this scenario, since the comparison baselines all utilize a 2-layer GNN, they are unaffected by 2-hop starved nodes and only minimally affected by 1-hop starved nodes. Consequently, our methods result in only slight improvements over the baselines on the Cora140 and Citeseer120 datasets. On the other hand, when $\tau = 20$, there are still $12,843$ 2-hop starved nodes present on the Pubmed dataset. Since we effectively eliminate these starved nodes without requiring additional graph convolutional layers, our methods can provide notable benefits on this dataset with an extremely low labeling rate.

Table 3: Test accuracy (%) of our proposed CUR extensions for GCN+CNN, GRCN, and SLAPS when eliminating different number of staved nodes on the Pubmed dataset.

| Staved nodes (%) | 100 | 80 | 60 | 40 | 20 | 10 | 0 |
|---|---|---|---|---|---|---|---|
| GCN+KNN_U | $68.66 \pm 0.05$ | $69.42 \pm 0.12$ | $70.20 \pm 0.22$ | $71.52 \pm 0.15$ | $72.40 \pm 0.27$ | $73.04 \pm 0.29$ | $74.12 \pm 0.32$ |
| GCN+KNN_R | $69.12 \pm 0.33$ | $69.46 \pm 0.22$ | $70.46 \pm 0.19$ | $71.90 \pm 0.15$ | $73.24 \pm 0.05$ | $74.00 \pm 0.14$ | $74.78 \pm 0.17$ |
| GRCN_U | $69.24 \pm 0.20$ | $72.14 \pm 0.35$ | $72.48 \pm 0.43$ | $73.04 \pm 0.52$ | $72.82 \pm 0.75$ | $72.96 \pm 0.87$ | $72.80 \pm 0.99$ |
| GRCN_R | $70.20 \pm 0.06$ | $72.34 \pm 0.30$ | $72.54 \pm 0.48$ | $72.98 \pm 0.37$ | $72.80 \pm 0.70$ | $72.80 \pm 0.74$ | $72.82 \pm 1.03$ |
| SLAPS_U | $74.86 \pm 0.79$ | $76.26 \pm 0.62$ | $76.48 \pm 0.61$ | $76.36 \pm 0.39$ | $76.48 \pm 0.67$ | $76.32 \pm 0.71$ | $76.74 \pm 0.59$ |
| SLAPS_R | $75.64 \pm 0.45$ | $76.44 \pm 1.27$ | $76.52 \pm 0.18$ | $76.50 \pm 1.22$ | $76.38 \pm 0.45$ | $76.70 \pm 0.59$ | $77.12 \pm 0.77$ |

Table 4: Parameter sensitivity of $\tau$ when applying our proposed method to GCN+CNN, GRCN, and SLAPS on the Pubmed dataset. The baseline results indicate the accuracy of the original methods.

| The value of $\tau$ | baseline (0) | 10 | 15 | 20 | 25 | 30 | 50 |
|---|---|---|---|---|---|---|---|
| GCN+KNN_U | $68.66 \pm 0.05$ | $71.86 \pm 0.26$ | $72.92 \pm 0.17$ | $73.50 \pm 0.31$ | $73.88 \pm 0.28$ | $74.12 \pm 0.32$ | $73.76 \pm 0.29$ |
| GCN+KNN_R | $68.66 \pm 0.05$ | $73.10 \pm 0.09$ | $73.90 \pm 0.14$ | $74.02 \pm 0.10$ | $74.56 \pm 0.12$ | $74.78 \pm 0.17$ | $74.62 \pm 0.12$ |
| GRCN_U | $69.24 \pm 0.20$ | $71.92 \pm 0.87$ | $72.56 \pm 0.77$ | $72.64 \pm 1.03$ | $72.80 \pm 0.99$ | $72.56 \pm 1.02$ | $71.80 \pm 1.13$ |
| GRCN_R | $69.24 \pm 0.20$ | $72.12 \pm 0.86$ | $72.54 \pm 0.77$ | $72.80 \pm 1.05$ | $72.82 \pm 1.03$ | $72.44 \pm 1.07$ | $71.86 \pm 1.11$ |
| SLAPS_U | $74.86 \pm 0.79$ | $75.98 \pm 1.12$ | $76.20 \pm 0.87$ | $75.58 \pm 0.90$ | $76.28 \pm 0.61$ | $76.74 \pm 0.59$ | $75.98 \pm 0.89$ |
| SLAPS_R | $74.86 \pm 0.79$ | $76.00 \pm 1.17$ | $76.48 \pm 0.69$ | $76.40 \pm 0.48$ | $77.12 \pm 0.77$ | $76.58 \pm 0.33$ | $76.50 \pm 0.59$ |

Table 5: Parameter sensitivity of $\alpha$ when applying our proposed method to GCN+CNN, GRCN, and SLAPS on the Pubmed dataset. The baseline results indicate the accuracy of the original methods.

| The value of $\alpha$ | baseline (0) | 0.01 | 0.1 | 1.0 | 10 | 50 | 100 |
|---|---|---|---|---|---|---|---|
| GCN+KNN_U | $68.66 \pm 0.05$ | $68.04 \pm 0.05$ | $67.96 \pm 0.08$ | $68.14 \pm 0.10$ | $69.92 \pm 0.12$ | $72.70 \pm 0.06$ | $74.12 \pm 0.32$ |
| GCN+KNN_R | $68.66 \pm 0.05$ | $67.96 \pm 0.10$ | $67.90 \pm 0.11$ | $68.18 \pm 0.07$ | $70.04 \pm 0.16$ | $73.22 \pm 0.07$ | $74.78 \pm 0.17$ |
| GRCN_U | $69.24 \pm 0.20$ | $69.24 \pm 0.10$ | $69.72 \pm 0.19$ | $71.92 \pm 0.24$ | $72.80 \pm 0.99$ | $67.22 \pm 3.93$ | $59.44 \pm 6.74$ |
| GRCN_R | $69.24 \pm 0.20$ | $69.24 \pm 0.10$ | $69.66 \pm 0.22$ | $71.94 \pm 0.22$ | $72.82 \pm 1.03$ | $68.98 \pm 4.05$ | $60.34 \pm 6.50$ |
| SLAPS_U | $74.86 \pm 0.79$ | $74.88 \pm 0.90$ | $74.48 \pm 0.72$ | $74.94 \pm 0.81$ | $76.26 \pm 0.80$ | $76.74 \pm 0.59$ | $76.08 \pm 0.97$ |
| SLAPS_R | $74.86 \pm 0.79$ | $74.62 \pm 1.51$ | $74.32 \pm 0.83$ | $74.76 \pm 0.71$ | $76.22 \pm 0.55$ | $77.12 \pm 0.77$ | $76.74 \pm 0.68$ |

### 4.4 Ablation Study

In this subsection, we aim to explore and answer the following questions.

*How many starved nodes should be eliminated?* The results shown in Table 1 indicate that eliminating all starved nodes contributes to the performance improvement of the baselines. It is important to understand the relationship between the number of starved nodes removed and the corresponding performance improvement of the baselines. To investigate this, we randomly remain $10\%, 20\%, 40\%, 60\%, 80\%$ starved nodes and evaluate the performance of GCN+CNN, GRCN, and SLAPS accordingly. The results on the Pubmed dataset are summarized in Table 3. We find that, for GCN+KNN, a smaller number of starved nodes leads to higher testing accuracy. For GRCN and SLAPS, however, we need to check the number of starved nodes to obtain optimal performance.

*How $\tau$ affects the performance.* Table 4 shows that the selection of $\tau$ slightly differentiates the performance of our proposed methods. In general, a larger value of $\tau$ leads to a higher improvement in performance. However, when $\tau$ is set too large, such as 50, the performance starts to degrade. This degradation occurs due to the introduction of incorrect labels when $\tau$ exceeds a certain threshold.

*How $\alpha$ affects the performance.* Table 5 presents the sensitivity of parameter $\alpha$ on the Pubmed dataset. It is observed that a relatively larger value of $\alpha$, such as 10 or 50, leads to a significant improvement in performance. This finding further emphasizes the effectiveness of our proposed regularization methods in enhancing the performance of existing LGI models.

## 5 Related Work

**Latent Graph Inference.** Given only the node features of data, latent graph inference (LGI) aims to simultaneously learn the underlying graph structure and discriminative node representations from the features of data [10, 24, 39]. For example, Jiang *et al.* [15] propose to infer the graph structure by combining graph learning and graph convolution in a unified framework. Yang *et al.* [45] model the topology refinement as a label propagation process. Jin *et al.* [16] explore some intrinsic properties of the latent graph and propose a robust LGI framework to defend adversarial attacks on graphs. Though effective, these methods require a prior graph to guide the graph inference process. Controversially,

some methods have been proposed to directly infer an optimal graph from the data. For example, Franceschi*et al.* [11] regard the LGI problem as a bilevel program task and learn a discrete probability distribution on the edges of the latent graph. Norcliffe-Brown *et al.* [29] focus on visual question answering task and propose to learn an adjacency matrix from image objects so that each edge is conditioned on the questions. Fatemi *et al.* [10] propose a self-supervision guided LGI method, called SLAPS, which yields supplementary supervision from node features through an additional self-supervision task. Our method is totally different from SLAPS as we provide supplementary supervision directly from the true labels. More importantly, our method is model-agnostic and can be easily integrated into most existing LGI methods.

**CUR Matrix Decomposition.** The CUR decomposition [2, 3] of a matrix $\mathbf{Q} \in \mathbb{R}^{n \times m}$ aims to find a column matrix $\mathbf{C} \in \mathbb{R}^{n \times c}$ with a subset of $c < m$ columns of $\mathbf{Q}$, and a row matrix $\mathbf{R} \in \mathbb{R}^{r \times m}$ with a subset of $r < n$ rows of $\mathbf{Q}$, as well as an intersection matrix $\mathbf{U} \in \mathbb{R}^{c \times r}$ such that the matrix multiplication of $\mathbf{CUR}$ approximates $\mathbf{Q}$. Unlike the SVD decomposition of $\mathbf{Q}$, the CUR decomposition obtains actual columns and rows of $\mathbf{Q}$, which makes it useful in various applications [20, 28]. In our method, instead of seeking an optimal matrix approximation, we employ CUR decomposition to identify and eliminate the starved nodes. The extracted intersection matrix $\mathbf{U}$ is then reconstructed and served as a regularization term to provide supplementary supervision for better latent graph inference. To the best of our knowledge, we are the first to introduce CUR matrix decomposition into the field of graph neural networks.

## 6    Conclusion

In this paper, we analyze the common problem of supervision starvation (SS) in existing latent graph inference (LGI) methods. Our analysis reveals that this problem arises due to the graph sparsification operation, which destroys numerous important connections between pivotal nodes and labeled ones. Building upon this observation, we propose to recover the corrupted connections and replenish the missed supervision for improved graph inference. To this end, we begin by defining $k$-hop starved nodes and transform the SS problem into a more manageable task of reducing starved nodes. Then, we present two simple yet effective solutions to identify the starved nodes, including a more efficient method inspired by CUR matrix decomposition. Subsequently, we eliminate the starved nodes by constructing and incorporating a regularization graph. In addition, we propose two straightforward strategies to tackle the potential issue known as weight contribution rate decay. Extensive experiments conducted on representative benchmarks demonstrate that our proposed methods consistently enhance the performance of state-of-the-art LGI models, particularly under extremely limited supervision.

## 7    Acknowledgments and Disclosure of Funding

We are very grateful to Bahare Fatemi for her valuable discussion of our work. We thank the anonymous NeurIPS reviewers for providing us with constructive suggestions to improve our paper. This material is based upon work supported by the Air Force Office of Scientific Research under award number FA9550-23-1-0290.

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

# Appendix

## A Proof of Theorem

### A.1 Proof of Theorem 1

*Proof.* To simplify the proof, let us assume that $\mathbf{A} = \mathbb{1}_{\mathbb{R}+}(\mathbf{A}) \in \{0,1\}^{n \times n}$[6]. Consider node $V_i$. If $\mathbf{A}_{ij} = 1$, it implies that node $V_j$ is one of the 1-hop neighbors of $V_i$. Now, if $(\mathbf{A}^2)_{ij} = \mathbf{A}_{i:} \cdot \mathbf{A}_{:j} = 1$, it means that there exists an 1-hop neighbor of $V_i$ that is also the 1-hop neighbor of $V_j$, indicating that node $V_j$ is one of the 2-hop neighbors of $V_i$. Similarly, if $(\mathbf{A}^3)_{ij} = (\mathbf{A})_{i:}^2 \cdot \mathbf{A}_{:j} = 1$, it signifies that there exists a 2-hop neighbor of $V_i$ that is also the 1-hop neighbor of $V_j$, implying that node $V_j$ is one of the 3-hop neighbors of $V_i$. By extension, if $(\mathbf{A}^k)_{ij} = 1$, it implies that node $V_j$ is the one of $k$-hop neighbors of $V_i$. Based on this observation, for $\forall j \in \{j \mid (\mathbf{A})_{ij} = 1 \cup (\mathbf{A}^2)_{ij} = 1 \cup \ldots \cup (\mathbf{A}^k)_{ij} = 1\}$, if node $V_j$ is unlabeled, it indicates that all the $\kappa$-hop neighbors of $V_i$ for $\kappa \in \{1, \ldots, k\}$ are unlabeled nodes. Therefore, according to Definition 1, node $V_i$ qualifies as a $k$-hop starved node. $\square$

**Illustration of Theorem 1.** To provide a clearer understanding, we present an example to illustrate the process of identifying $k$-hop starved nodes based on the given adjacency matrix. Fig. 3 depicts a graph consisting of 6 nodes, with 2 labeled and 4 unlabeled. The edges between the nodes are shown in the figure, with all edge weights set to 1 for simplicity. To identify the $k$-hop starved nodes, we need to determine the $k$-hop neighbors for each node. The steps below demonstrate the identification process of $k$-hop neighbors based on the given graph in Fig. 3, where the $k$-hop neighbors for each node are listed at the end of each row of the corresponding matrices:

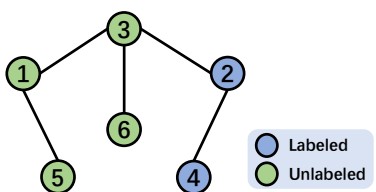

Figure 3: A simple graph consisting of 6 nodes with 2 labeled and 4 unlabeled.

$$
\text{Identifying 1-hop neighbors based on } \mathbf{A}: \quad
\begin{array}{c}
1 \\ 2 \\ 3 \\ 4 \\ 5 \\ 6
\end{array}
\begin{array}{cccccc}
1 & 2 & 3 & 4 & 5 & 6 \\
\end{array}
\left[
\begin{array}{cccccc}
1 & 0 & 1 & 0 & 1 & 0 \\
0 & 1 & 1 & 1 & 0 & 0 \\
1 & 1 & 1 & 0 & 0 & 1 \\
0 & 1 & 0 & 1 & 0 & 0 \\
1 & 0 & 0 & 0 & 1 & 0 \\
0 & 0 & 1 & 0 & 0 & 1 \\
\end{array}
\right]
\begin{array}{l}
(V_3, V_5) \\
(V_3, V_4) \\
(V_1, V_2, V_6) \\
(V_2) \\
(V_1) \\
(V_3)
\end{array}
$$

Since nodes $V_2$ and $V_4$ are labeled, we identify the 1-hop starved nodes as $\{V_1, V_5, V_6\}$.

$$
\text{Identifying 2-hop neighbors based on } \mathbf{A}^2: \quad
\begin{array}{c}
1 \\ 2 \\ 3 \\ 4 \\ 5 \\ 6
\end{array}
\begin{array}{cccccc}
1 & 2 & 3 & 4 & 5 & 6 \\
\end{array}
\left[
\begin{array}{cccccc}
3 & 1 & 2 & 0 & 2 & 1 \\
1 & 3 & 2 & 2 & 0 & 1 \\
2 & 2 & 4 & 1 & 1 & 2 \\
0 & 2 & 1 & 2 & 0 & 0 \\
2 & 0 & 1 & 0 & 2 & 0 \\
1 & 1 & 2 & 0 & 0 & 2 \\
\end{array}
\right]
\begin{array}{l}
(V_2, V_6) \\
(V_1, V_6) \\
(V_4, V_5) \\
(V_3) \\
(V_3) \\
(V_1, V_2)
\end{array}
$$

Now, we can identify 2-hop starved nodes from the set $\{V_1, V_5, V_6\}$ as $\{V_5\}$.

$$
\text{Identifying 3-hop neighbors based on } \mathbf{A}^3: \quad
\begin{array}{c}
1 \\ 2 \\ 3 \\ 4 \\ 5 \\ 6
\end{array}
\begin{array}{cccccc}
1 & 2 & 3 & 4 & 5 & 6 \\
\end{array}
\left[
\begin{array}{cccccc}
7 & 3 & 7 & 1 & 5 & 3 \\
3 & 7 & 7 & 5 & 1 & 3 \\
7 & 7 & 10 & 3 & 3 & 6 \\
1 & 5 & 3 & 4 & 0 & 1 \\
5 & 1 & 3 & 0 & 4 & 1 \\
3 & 3 & 6 & 1 & 1 & 4 \\
\end{array}
\right]
\begin{array}{l}
(V_4) \\
(V_5) \\
(\varnothing) \\
(V_1, V_6) \\
(V_2, V_6) \\
(V_4, V_5)
\end{array}
$$

---
[6]Note that, applying the function $\mathbb{1}_{\mathbb{R}+}(\cdot)$ to $\mathbf{A}$ does not affect the proof process.

We observe that there are no 3-hop starved nodes.

$$
\text{Identifying 4-hop neighbors based on } \mathbf{A}^4 : \quad
\begin{array}{c}
1 \\ 2 \\ 3 \\ 4 \\ 5 \\ 6
\end{array}
\begin{array}{c}
\begin{array}{cccccc}
1 & 2 & 3 & 4 & 5 & 6
\end{array} \\
\left[
\begin{array}{cccccc}
19 & 11 & 20 & 4 & 12 & 10 \\
11 & 19 & 20 & 12 & 4 & 10 \\
20 & 20 & 30 & 10 & 10 & 16 \\
4 & 12 & 10 & 9 & 1 & 4 \\
12 & 4 & 10 & 1 & 9 & 4 \\
10 & 10 & 16 & 4 & 4 & 10
\end{array}
\right]
\end{array}
\begin{array}{c}
(\varnothing) \\ (\varnothing) \\ (\varnothing) \\ (V_5) \\ (V_4) \\ (\varnothing)
\end{array}
$$

We observe that there are no 4-hop starved nodes.

Note that, a node identified as $k$-hop starved node is also considered as a $(k-1)$-hop starved node, as exemplified by node $V_5$. Consequently, if there are no $k$-hop starved nodes present, it follows that there are no $(k+1)$-hop starved nodes.

## A.2 Proof of Theorem 2

*Proof.* Given that $\mathbf{C} = \mathbf{A}[:, col\_mask] \in \mathbb{R}^{n \times c}$, $\mathbf{C}$ models the affinities between all nodes and $c$ labeled nodes. Consequently, if $(row\_mask)_i = \mathbb{1}_{\mathbb{R}^-}(\mathbf{C}\mathbb{1}_c)_i = 1$, indicating that $(\mathbf{C}\mathbb{1}_c)_i = 0$, we can deduce that there are no labeled 1-hop neighbors of node $V_i$. As a result, node $V_i$ belongs to the set of 1-hop starved nodes, denoted as $\mathtt{Set}_1(r) = \{V_i | i \in \mathtt{RM}_+\}$. Additionally, it follows that $\mathbf{U} = \mathbf{A}[row\_mask, col\_mask] = \mathbf{0} \in \mathbb{R}^{r \times c}$. Considering $\mathbf{R} = \mathbf{A}[row\_mask, :] \in \mathbb{R}^{r \times n}$, where $\mathbf{R}$ models the affinities between the $r$ 1-hop starved nodes and all nodes, we can examine each $i \in \mathtt{RM}_+$ (where $\mathtt{RM}_+$ denotes the indexes of all 1-hop starved nodes). If all 1-hop neighbors of $V_i$ (*i.e.,* $\forall j$ satisfying $[\mathbb{1}_{\mathbb{R}^+}(\mathbf{R})]_{ij} = 1$) are 1-hop starved nodes (*i.e.,* $j \in \mathtt{RM}_+$), then node $V_i$ qualifies as a 2-hop starved node. $\qquad\square$

**Illustration of Theorem 2.** To facilitate understanding, we continue using Fig. 3 as an example for illustration. According to Theorem 2, the corresponding $\mathbf{C}, \mathbf{U}, \mathbf{R}$ matrices are as follows:

$$
\mathbf{C} : \quad
\begin{array}{c}
1 \\ 2 \\ 3 \\ 4 \\ 5 \\ 6
\end{array}
\begin{array}{c}
\begin{array}{cc} 2 & 4 \end{array} \\
\left[
\begin{array}{cc}
0 & 0 \\
1 & 1 \\
1 & 0 \\
1 & 1 \\
0 & 0 \\
0 & 0
\end{array}
\right]
\end{array}
\quad ; \quad
\mathbf{R} : \quad
\begin{array}{c}
1 \\ 5 \\ 6
\end{array}
\begin{array}{c}
\begin{array}{cccccc} 1 & 2 & 3 & 4 & 5 & 6 \end{array} \\
\left[
\begin{array}{cccccc}
1 & 0 & 1 & 0 & 1 & 0 \\
1 & 0 & 0 & 0 & 1 & 0 \\
0 & 0 & 1 & 0 & 0 & 1
\end{array}
\right]
\end{array}
\quad ; \quad
\mathbf{U} : \quad
\begin{array}{c}
1 \\ 5 \\ 6
\end{array}
\begin{array}{c}
\begin{array}{cc} 2 & 4 \end{array} \\
\left[
\begin{array}{cc}
0 & 0 \\
0 & 0 \\
0 & 0
\end{array}
\right]
\end{array}
$$

Based on the $\mathbf{C}, \mathbf{U}, \mathbf{R}$ matrices, we can determine that $row\_mask = [1,0,0,0,1,1]^\top$, $\mathtt{RM}_+ = \{1,5,6\}$, the 1-hop starved nodes are $V_1, V_5, V_6$, and the 2-hop starved node is $V_5$.

## A.3 Proof of Proposition 1

*Proof.* Recall that $\widehat{\rho}_+ = \frac{\sum_{i \in \mathtt{RM}_+} \alpha \widehat{\mathbf{U}}_{ij}}{\sum_i \widetilde{\mathbf{A}}_{ij}}$ and $\rho_+ = \frac{\sum_{i \in \mathtt{RM}_+} \alpha \widetilde{\mathbf{U}}_{ij}}{\sum_i \widetilde{\mathbf{A}}_{ij}}$. For $\forall j \in \mathtt{CM}_+$, if we randomly select $\tau$ out of $c$ labeled nodes as the supplementary adjacent points for each 1-hop starved node $V_i$, then the probability of $\widehat{\mathbf{U}}_{ij} > 0$ is $\frac{\tau}{c}$. Considering the $j$-th column ($j \in \mathtt{CM}_+$), if $\sum_{i \in \mathtt{RM}_+} \alpha \widehat{\mathbf{U}}_{ij} = \sum_{i \in \mathtt{RM}_+} \alpha \widetilde{\mathbf{U}}_{ij}$ (*i.e.,* $\widehat{\rho}_+ = \rho_+$ ), it means that all the 1-hop starved nodes $V_i$ (for $i \in \mathtt{RM}_+$) select $V_j$ as the supplementary adjacent point. Since there are $r$ 1-hop starved nodes, the probability of $\widehat{\rho}_+ = \rho_+$ is $\left(\frac{\tau}{c}\right)^r$. Moreover, since $\tau < c$, we have $\widehat{\rho}_+ = \frac{\sum_{i \in \mathtt{RM}_+} \alpha \widehat{\mathbf{U}}_{ij}}{\sum_i \widetilde{\mathbf{A}}_{ij}} \leq \rho_+$, where $\widehat{\rho}_+ = \rho_+$ with only a probability of $\left(\frac{\tau}{c}\right)^r$. $\qquad\square$

## A.4 Proof of Proposition 2

*Proof.* Recall that $\widehat{\rho}_- = \frac{\sum_{i \notin \mathtt{RM}_+} \widetilde{\mathbf{A}}_{ij} + \alpha \sum_{i \notin \mathtt{RM}_+} \mathbf{Q}_{ij}}{\sum_i \widetilde{\mathbf{A}}_{ij} + \alpha \sum_{i \notin \mathtt{RM}_+} \mathbf{Q}_{ij}}$ and $\rho_- = \frac{\sum_{i \notin \mathtt{RM}_+} \widetilde{\mathbf{A}}_{ij}}{\sum_i \widetilde{\mathbf{A}}_{ij}}$. For $\forall j \in \mathtt{CM}_+$, if we randomly select $\tau$ out of $c$ labeled nodes as the supplementary adjacent points for each non-starved

Table 6: Detailed description of the datasets used in experiments.

| Dataset / Statistic | Nodes | Edges | Features | Classes | Labeling Rate |
|---|---|---|---|---|---|
| ogbn-arxiv | 169,343 | 1,166,243 | 128 | 40 | 53.70% |
| Cora140 | 2,708 | 5,429 | 1,433 | 7 | 5.17% |
| Cora390 | 2,708 | 5,429 | 1,433 | 7 | 14.40% |
| Citeseer120 | 3,327 | 4,732 | 3,703 | 6 | 3.61% |
| Citeseer370 | 3,327 | 4,732 | 3,703 | 6 | 11.12% |
| Pubmed | 19,717 | 44,338 | 500 | 3 | 0.30% |

node, then the probability of $\mathbf{Q}_{ij} = 0$ is $\left(1 - \frac{\tau}{c}\right)$. Considering the $j$-th column ($j \in \mathtt{CM}_+$), if $\alpha \sum_{i \notin \mathtt{RM}_+} \mathbf{Q}_{ij} = 0$ (*i.e.*, $\widehat{\rho}_- = \rho_-$), it means that all the non-starved nodes $V_i$ (for $i \notin \mathtt{RM}_+$) do not select $V_j$ as the supplementary adjacent point. Since there are $r$ 1-hop starved nodes, the probability of $\widehat{\rho}_- = \rho_-$ is $\left(1 - \frac{\tau}{c}\right)^{n-r}$. Moreover, since $\sum_i \widetilde{\mathbf{A}}_{ij} > \sum_{i \notin \mathtt{RM}_+} \widetilde{\mathbf{A}}_{ij}$ ($|\mathtt{RM}_+| = r$), we have $\widehat{\rho}_- \geq \rho_-$, where $\widehat{\rho}_- = \rho_-$ with only a probability of $\left(1 - \frac{\tau}{c}\right)^{n-r}$. $\qquad\square$

# B Experimental Settings

## B.1 Dataset Description

Table 6 provides a comprehensive overview of the datasets used in our experiments, including various statistical characteristics. Please refer to the table for detailed information regarding the datasets. Note that, the original edge features are not used in the experiments as our goal is to infer a latent graph from the node features of the datasets.

## B.2 Baselines

In our experiments, we utilize publicly available code repositories provided by the respective authors and follow the hyperparameter settings outlined in their papers [4, 10, 13, 19, 46]. We rerun the codes for all the methods and record the best testing accuracy. Below, we summarize the implementation details and provide the code links for each of the used methods:

**GCN+KNN** [19]: https://github.com/tkipf/pygcn.

**GCN&KNN** [19]: https://github.com/tkipf/pygcn.

For GCN&KNN, we implement the codes by ourselves. For simplicity, we adopt the following Dirichlet energy [4] as the graph regularization loss:

$$\mathcal{L}_{\text{reg}} = \frac{1}{2n^2} \sum_{i=1}^{n} \sum_{j=1}^{n} \widehat{\mathbf{A}}_{ij} ||\mathbf{X}_{i:} - \mathbf{X}_{j:}||^2.$$

**IDGL** [4]: https://github.com/hugochan/IDGL.

**GRCN** [46]: https://github.com/PlusRoss/GRCN.

**SLAPS** [10]: https://github.com/BorealisAI/SLAPS-GNN.

Note that, SLAPS is a multi-task-based approach that involves training an additional self-supervision task for latent graph inference. To ensure computational efficiency, we set the maximum number of epochs to 1000 for ogbn-arxiv dataset. We adopt the following denoising autoencoder loss as the graph regularization loss [10]:

$$\mathcal{L}_{\text{reg}} = F(\mathbf{X}_{idx}, \mathtt{GNN}_{\mathtt{DAE}}(\widetilde{\mathbf{X}}, A; \theta_{\mathtt{GNN}_{\mathtt{DAE}}})_{idx}).$$

where $idx$ are the selected indices, and $F$ is the binary cross-entropy loss or the mean-squared error loss [10].

**LCGS** [13]: https://github.com/hu-my/LCGS.

Table 7: Accuracy, parameters and FLOPs of SLAPS when using a different number of GNN layers.

| Layers | Accuracy | Parameters | FLOPs |
|---|---|---|---|
| 2 | $74.86 \pm 0.79$ | 645.76K | 12.70G |
| 4 | $70.24 \pm 1.58$ | 680.90K | 13.39G |
| 8 | $41.18 \pm 0.88$ | 751.17K | 14.76G |

## C Further Discussions

### C.1 Deeper GNNs

*Why using deeper GNNs is not an optimal solution?* This is because using deeper will bring up new issues as illustrated below. **First**, although increasing GNN to 4 layers reduces the number of starved nodes from near $3,000$ to near $500$ for the Citeseer120 dataset, the number of starved nodes still accounts for nearly $15\%$. In fact, the number of starved nodes depends on several factors, including the labeling rate of nodes, the graph sparsification process, and so on. *How to select the number of layers of GNNs to reduce the starved nodes for different datasets and different LGI methods is not easy*. **Second**, deeper GNNs typically provide inferior performance. To allow a $k$-hop starved node to receive information from labeled nodes, the GNNs need to have at least $k + 1$ layers. However, as the number of layers $k$ increases, the GNNs may still fail to propagate supervision information from labeled nodes to $k$-hop starved nodes due to the over-squashing issue [1]. Moreover, deeper GNNs will make the oversmoothing problem [10, 21] be even more severe. Empirically, we test the SLAPS method on the Pubmed dataset using a different number of layers, and the corresponding accuracies can be seen in Table 7. As we can see, the accuracy of SLAPS with a 8-layer GNN is extremely low, demonstrating that *the generalization of GNNs cannot be guaranteed by simply increasing the number of layers*. **Moreover**, deeper GNNs make the model more complexity. Table 7 lists the number of parameters and the float point operations (FLOPs) of SLAPS method with 2, 4, and 8-layer GNNs. We observe that *as the number of layers increases, both the number of parameters and the FLOPs increase*. In fact, the majority of existing LGI methods typically only adopt 2-layer GNNs for effectiveness and efficiency.

### C.2 Degree Distribution

*Why introducing connections between labeled and unlabeled nodes helps the models?* We provide a basic insight to answer this question. As we know, an unlabeled node must belong to a specific class, and in the node classification task, a specific class must have some labeled nodes. As a result, we can derive the prior assumption that a good classification model will make an unlabeled node closer to some labeled nodes of the same class. That is why we select the closest labeled nodes as the supplementary adjacent points for the starved nodes. If we compulsorily add such connections between the unlabeled node and its closest labeled nodes, the prior information will be exploited during training, helping the models work better. It is worth noting that introducing connections between labeled and unlabeled nodes can lead to a substantial rise in the degrees of labeled nodes. A potential question is that whether altering the distribution of node degrees will have an adverse effect on the model. To be honest, we have not yet observed any adverse effects from the current experimental results. Nevertheless, it is still worth exploring whether the divergence in degree distributions will potentially hinder the model's performance. What the exact relationship is between the degree distributions and the model's performance in latent graph inference task is a difficult and open question that deserves in-depth exploration.

### C.3 Definition of Regularization Graph

*Why selecting $k = 1$ to define the regularization graph between $k$-hop starved nodes and labeled ones?* In fact, we can set any value for $k$ to define the regularization graph. However, if we set $k > 1$, there will be a potential issue. For illustration, let us set $k = 3$ and eliminate the 3-hop starved nodes by adding the regularization graph. In this case, 1-hop and 2-hop starved nodes will still exist since only 3-hop starved nodes are removed. To address this, a simple solution is to directly set $k = 1$ to define the regularization graph. Why only setting $k = 1$ works? According to the definition of starved nodes, we know that a $k$-hop starved node also qualifies as a $(k - 1)$-hop starved node. Therefore, if

Table 8: Test accuracy (%) of different models corresponding to the best validation performance.

| GCN+KNN | $67.28 \pm 0.38$ | GRCN | $68.62 \pm 0.19$ | SLAPS | $74.50 \pm 0.33$ |
|---------|------------------|------|------------------|-------|------------------|
| GCN+KNN_U | $72.76 \pm 0.29$ | GRCN_U | $72.30 \pm 0.95$ | SLAPS_U | $76.00 \pm 0.41$ |
| GCN+KNN_R | $72.98 \pm 0.31$ | GRCN_R | $72.14 \pm 0.79$ | SLAPS_R | $76.86 \pm 0.83$ |

we eliminate 1-hop starved nodes then all $m$-hop starved nodes for $m > 1$ will be eliminated. Due to the effectiveness and simplicity, we therefore select $k = 1$.

### C.4 Generalization Performance

To show the results of the models corresponding to the best validation performance, we conduct more experiments on the Pubmed dataset, and the experimental results are listed in Table 8. As we can see, our methods still improve the baselines, which further indicates their effectiveness.

## D Limitation

According to Theorem 1, we can identify $k$-hop starved nodes for $k \geq 1$. Additionally, based on Theorem 2 and CUR matrix decomposition, we demonstrate the identification of $k$-hop starved nodes for $k \in \{1, 2\}$. How to identify $k$-hop starved nodes for $k > 2$ based on CUR decomposition remains an interesting direction for further investigation. Nevertheless, it is worth noting that by eliminating all $\kappa$-hop starved nodes, there will be no $k$-hop starved nodes for $\forall k > \kappa$. Consequently, we think that only identifying $k$-hop starved nodes for $k \in \{1, 2\}$ is sufficient and efficient to address the problem of supervision starvation.

