# OpenReview forum: "Latent Graph Inference with Limited Supervision"
_NeurIPS.cc/2023/Conference — NeurIPS 2023 poster_

### Official Review · Reviewer_6cu2 · 2023-06-23

**Soundness:** 4 excellent
**Presentation:** 4 excellent
**Contribution:** 4 excellent
**Rating:** 7
**Confidence:** 3

**Summary:**

The authors figure out that the graph sparsification operation results in the supervision starvation problem in latent graph inference (LGI). They propose to identify k-hop starved nodes and diminish the starved nodes by incorporating a regularization adjacency matrix into the initial one. They further reduce the computational cost by using CUR matrix decomposition and tackle the weight contribution rate decay problem via some simple strategies. The effectiveness of the proposed method is validated on well-known benchmarks.

Update after rebuttal:
The authors have addressed my concern through the rebuttal. The score remains unchanged.

**Strengths:**

1. This paper is well-written and easy to follow.
2. This paper identifies the supervision starvation problem for LGI.
3. The proposed method is well-motivated and theoretically justified.
4. The proposed method is model-agnostic and can be seamlessly integrated into various LGI models.
5. The empirical results are significant on benchmarks.

**Weaknesses:**

From the analysis in Section 4.2, the proposed method may be sensitive to the hyper-paramters $\tau$ and $\alpha$.

**Questions:**

In Eq. (4), the refined matrix $\widetilde{\mathbf{A}}$ becomes a weighted adjacency matrix when the positive parameter $\alpha \neq 1$. Will tuning $\alpha$ help with methods developed for unweighted graphs?

---

> ### Author Rebuttal · Authors · 2023-08-09
>
> **We really appreciate that the Reviewer identifies our contributions and provides constructive comments. We address your concerns as below**.
>
> **W 1:  Parameter sensitivity**.
> Thank you for this valuable comment. In Sec. 4.2, we discussed how $\tau$ and $\alpha$ affect performance. In our method, $\tau$ determines the number of labeled nodes that will be connected by each starved node, and $\alpha$ balances the contribution between the initial adjacency matrix and the regularization one. As we can see from Table 3, our proposed method is not very sensitive to $\tau$. For different baselines, our method can obtain the best performance when $\tau$ belongs to $\lbrace 25, 30 \rbrace$. Therefore, we can set the same $\tau$ for different baselines. From Table 4, we found that the performance is relatively sensitive to $\alpha$. In fact, this is because the value of $\alpha$ depends more on the baseline we used. For GCN+KNN, a larger $\alpha$ leads to better performance. The potential reason is that the pre-constructed graph in GCN+KNN cannot be optimized. Therefore, enlarging $\alpha$ will make the regularization graph contribute more to the final loss and achieve more improvement. For GRCN and SLAPS, since the initial graph is revised by node embeddings or updated by self-supervision signals, a moderate value of $\alpha$ needs to be set to achieve a balance between the initial graph and the regularization one. We will add more discussion on this point in the final version.
>
> **Q 1:  Unweighted graphs**.
> Thank you for asking this constructive question. In fact, all the latent graph inference methods generate weighted graphs with real-valued edge weights (the matrix $A$ is real-valued) from their graph generators. Therefore, $\widetilde A$ is always a weighted adjacency matrix, no matter how we set the value of $\alpha$. The values of edge weights are basically calculated based on different distance functions designed by different baselines. Typically, the smaller the distance between nodes, the larger the corresponding edge weights. To test whether turning $\alpha$ helps for unweighted graphs, we use SLAPS as an example to conduct more experiments. Specifically, we change the original graph generator of SLPAS to generate an unweighted graph and test its performance on the Pubmed dataset. Then, we tune the value of $\alpha$ and test the corresponding performance of the SLPAS_U and SLAPS_R methods. The experimental results are listed below. As we can see, for unweighted graphs, our method can still improve the baseline of SLPAS. However, the performance improvement is not sensitive to the value of $\alpha$ since the learned graphs are unweighted.
>
> | **$\alpha$** | **baseline(0)** | **0.01** | **0.1** | **1.0** | **10** | **50** | **100** |
> |-------|-------------|------|---------|------|----|----|-----|
> | **SLAPS_U** | 66.02 ± 1.47 | 68.26 ± 1.23 | 68.68 ± 1.04 | 68.72 ± 1.30 | 68.72 ± 1.30 | 68.72 ± 1.30 | 68.72 ± 1.30 |
> | **SLAPS_R** | 66.02 ± 1.47 | 68.28 ± 1.20 | 68.34 ± 0.85 | 68.64 ± 1.21 | 68.64 ± 1.21 | 68.64 ± 1.21 | 68.64 ± 1.21 |

---

> > ### Comment · Reviewer_6cu2 · 2023-08-11
> >
> > Thanks for your further analysis of parameter sensitivity and additional experiments on unweighted graphs. I have no further questions.

---

> > > ### Author Response · Authors · 2023-08-11
> > > **Thank you!**
> > >
> > > Thank you very much for your quick response and valuable comments. We will try our best to revise our manuscript accordingly. Aagin, thank you so much for helping us improve the paper!

---

### Official Review · Reviewer_se4r · 2023-07-04

**Soundness:** 3 good
**Presentation:** 3 good
**Contribution:** 3 good
**Rating:** 7
**Confidence:** 4

**Summary:**

The paper points common LGI methods suffer from the issue of `supervision starvation`.
It also observes this issue is actually caused by the graph sparsification operation.
To address this problem, the paper proposes to restore the corrupted affinities and replenish the missed supervision.
It presents the `CUR matrix decomposition` to reduce the computational burden and eliminates the starved nodes by reconstructing the destroyed connections
The method is model-agnostic and can be seamlessly integrated into existing LGI methods.
Extensive experiments show promising results.

**Strengths:**

1) Originality.
The identifies graph sparsification is the main cause of the supervision starvation (SS) in LGI.
The paper is solid in analyzing problems with insights.

2) Quality.
This method is supported by both theoretical (Theorem 1, 2) and experimental aspects, having strong persuasiveness.

3) Clarity.
The paper is well written and organized.

4) Significance.
LGI is a common task graph learning. This method can be potentially applied in many situations.

**Weaknesses:**

1) The paper should clearly explain why using `CUR Decomposition` is more efficient.
The experimental results only show a higher accuracy, but does not reflect the efficiency.
So this advantage is not well supported.

2) The section3.2 and 3.3 are not easy to follow.
The reviewers and potential readers may prefer a popular version.


**Questions:**

1) The $L_{reg}$ should have a clear mathematical form in paper in Equal 9.

2) The authors are encouraged to discuss more about SoTA GNNs [15,23,37,40] since the proposed approach is model-agnostic.
The reviewer is curious why these methods were mentioned in Section1 but not compared in experiments.

**Limitations:**

The LGI is a common task in graph learning and all experiments are conducted on public and popular datasets.
The reviewer see no negative societal impact and limitations of this work.

---

> ### Author Rebuttal · Authors · 2023-08-09
>
> **We really appreciate that the Reviewer recognizes our contributions and originality, and gives us useful suggestions. We give our response below**.
>
> **W 1: Why CUR Decomposition is more efficient**.
> Thank you for this important comment. As stated in Lines 48-53, Sec. 1, when we say “more efficient”, we mean that using CUR Decomposition (the method proposed in Sec. 3.2) to identify the starved nodes is more efficient in comparison with the method based on the k-th power of a given adjacency matrix (the method proposed in Sec. 3.1). To support this point, we conduct more experiments to compare the time performance of these two methods. The experimental results on running time are listed below. As we can see, using CUR Decomposition is more efficient than the method based on the k-th power of the adjacency matrix, especially on the larger Pubmed dataset.
>
> | **Datasets** | **Cora140** | **Cora390** | **Citeseer120** | **Citeseer370** | **Pubmed** |
> |-----------|--------------|------------|---------------|--------------|----------------|
> | **k-th power of adjacency matrix**   |     1.10      |     1.05       |      1.65         |      1.67         |    296.81   |
> | **CUR Decomposition**                   |     0.21      |     0.22       |      0.35         |      0.36         |      11.68   |
>
> **W 2: Revise sections 3.2 and 3.3**.
> We thank the Reviewer for pointing this out. To better illustrate Theorem 2, we provide a simple example in the supplementary material. In addition, we will double polish these sections to make them easy to follow.
>
> **Q 1: Clear form of L_reg**.
> Thank you for this helpful suggestion. As stated in Sec. 3.4, different LGI methods adopt different graph regularization loss $L_{reg}$. For example, IDGL adopts the following Dirichlet energy as the graph regularization loss [4]:
> $$L_{reg} = \frac{1}{2n^2} \sum_{i,j}A_{ij}||x_i-x_j||^2 = \frac{1}{n^2} tr(X^TLX).$$
> SLAPS designs the following denoising autoencoder loss as the graph regularization loss [9]:
> $$L_{reg} = F(X_{idx}, GNN_{DAE}(\widetilde X, A; \theta_{GNN_{DAE}})_{idx}).$$
> We will add clear mathematical forms for these different regularization losses and give more discussion in the final version.
>
> **Q 2: Discuss more about SOTA GNNs**.
> We thank the Reviewer for this comment. We think [15][23] cannot be regarded as latent graph inference methods since they do not infer latent graphs solely based on the node features. More specifically, [15] requires a prior graph as the input and then modifies the prior graph structure based on the sparsity and low-rank assumptions. The main goal of this method is to tackle the adversarial attack problem rather than infer a good latent graph from scratch. [23] aims to solve the graph comparison problem based on graph kernels and graph neural networks. This method also does not involve the inference of latent graphs. Besides, [37] proposes a robust similarity measure based on the B-Attention mechanism for multiple clustering and ReID tasks, and [40] designs a Transformer model that scales all-pair message passing to large node classification graphs. These two methods do not involve the graph sparsification operation and thus do not encounter the supervision starvation problem. In summary, these methods [15, 23, 37, 40] are different from the latent graph inference methods used in our experiments. We will discuss more about these methods in the final version.

---

> > ### Comment · Reviewer_se4r · 2023-08-19
> > **Additional Comments**
> >
> > Thanks for your response.
> > I still believe that these SoTA methods should be compared,  at least one of them, instead of just talking.
> > At least an experiment result of them should be put on to show their shortcomings, Otherwise, I would think the experiment is insufficient.

---

> > > ### Author Response · Authors · 2023-08-19
> > > **Thanks for your reply!**
> > >
> > > Dear Reviewer se4r,
> > >
> > > Thank you very much for your reply and instructive suggestions. We are doing the experiments you requested now. We will try our best to show the experimental results before the discussion period.
> > >
> > > Thanks again for your help in improving our paper!
> > >
> > > Best regards,
> > >
> > > Authors

---

> > > ### Author Response · Authors · 2023-08-21
> > > **New experimental results.**
> > >
> > > Thank you again for your constructive suggestions. We add one more SOTA method, Pro-GNN [15], for experiments. We use their source codes and adopt the same data partitioning. In fact, Pro-GNN does not infer a latent graph from scratch based on the node features. It requires a prior semantic graph as the input. To align with our latent graph inference task, we construct a KNN graph as the prior graph for Pro-GNN. The experimental results on the Cora and Citeseer datasets are shown below (it cannot be trained on other datasets directly due to the out-of-memory issue). As we can see, our proposed methods still achieve better performance.
> > >
> > > | Datasets | Cora | Citeseer |
> > > | --- | --- | --- |
> > > | Pro-GNN                    | 70.73 ± 0.35 | 71.76 ± 0.14 |
> > > | Pro-GNN_U (ours)     | 70.76 ± 0.32 | 71.86 ± 0.95 |
> > > | Pro-GNN_R (ours)     | 70.85 ± 0.39 | 72.18 ± 0.33 |
> > >
> > > We hope this can address your concern. And again, thanks for helping us improve the quality of our paper.

---

### Official Review · Reviewer_tzHX · 2023-07-05

**Soundness:** 3 good
**Presentation:** 3 good
**Contribution:** 3 good
**Rating:** 6
**Confidence:** 5

**Summary:**

The paper proposes a method for latent graph inference (aka graph structure learning) based on the idea of mitigating the supervision starvation problem present when jointly learning the underlying graph structure and node representations. The paper claims that the supervision starvation problem is caused by the sparsifier operation present in many existing models in the literature. It further proposes to recover (some) starved edges to resolve the issue and shows that reducing the number of starved edges consistently improves the performance on multiple benchmarks.

**Strengths:**

The problem of latent graph inference and the supervision starvation problem present in many existing approaches are important directions that are recently gaining more and more attention. The paper proposes an approach to identify the starved edges and add them back to the graph structure that is backed up by CUR decomposition and theoretical results. The final analysis shows the proposed strategy improves the performance of state-of-the-art LGI methods. The method is shown to be more effective when supervision is less.

**Weaknesses:**

W1: One main weakness of the model is that the results are obtained using the average of the top five best testing performances. I understand that this setup was fixed across multiple baselines and datasets but to measure the generalization of the model, I would expect the results to be an average over testing performance of the models corresponding to the best validation performance. This is especially important as the charts in Figure 2 show a lot of instabilities in the test loss of multiple models.

W2: the proposed model adds some edges to the graph structure and compares itself to the base model without those edges. I suggest adding a baseline as the base model + random edges to make sure the performance obtained is not just because there are more edges present.

**Questions:**

Q1: Can you elaborate more on what models M_U and M_R are referring to? My understanding so far is that M_U is referring to Equation 6 and M_R is referring to Equation 8.

Q2: Can you explain what edge weights have been used on both existing and starved edges? Is this the real-valued edge from the output of the graph generator for both type of edges?

Q3: What type of graph generators has been used to obtain the results in Table 1?

**Limitations:**

One limitation of the model is that Theorem 2 only works for k of 1 and 2. For many academic datasets, using one or two layers of GNNs is enough. However, in many real-world applications, deeper GNNs are needed to send the long-range dependencies. The paper does not provide any insights on how the idea can be extended for higher values of ks.

---

> ### Author Rebuttal · Authors · 2023-08-09
>
> **We really appreciate the Reviewer’s valuable comments. We address your concerns as below.**
>
> **W 1:  Best validation performance**.
> We thank the Reviewer for this important suggestion, and we agree that the setting you mentioned is more reasonable. *To show the results of the models corresponding to the best validation performance, we conduct more experiments on the Pubmed dataset, and the experimental results are listed below*. As we can see, our methods still improve the baselines, which further indicates their effectiveness. Besides, we think the instability phenomenon depends on the baselines we used. For GRCN, its curves are stable. Although GCN+KNN and SLPAS show some instabilities in the loss and accuracy curves, *for all the methods, the testing accuracy curves of our models lie above the baselines*, further demonstrating the effectiveness of our models.
>
> The performance of the models corresponding to the best validation performance:
> |    |   |    |   |    |    |
> |---|---|---|---|---|---|
> | **GCN+KNN**      | 67.28 ± 0.38 | **GRCN**     | 68.62 ± 0.19 | **SLAPS**      | 74.50 ± 0.33  |
> | **GCN+KNN_U** | 72.76 ± 0.29 | **GRCN_U** | 72.30 ± 0.95 | **SLAPS_U** | 76.00 ± 0.41  |
> | **GCN+KNN_R** | 72.98 ± 0.31 | **GRCN_R** | 72.14 ± 0.79 | **SLAPS_R** | 76.86 ± 0.83  |
>
> The performance of the models corresponding to the best testing performance:
> |    |   |    |    |   |    |
> |---|---|---|---|---|---|
> | **GCN+KNN**      | 68.66 ± 0.05 | **GRCN**     | 69.24 ± 0.20 | **SLAPS**      | 74.86 ± 0.79 |
> | **GCN+KNN_U** | 74.12 ± 0.32 | **GRCN_U** | 72.80 ± 0.99 | **SLAPS_U** | 76.74 ± 0.59 |
> | **GCN+KNN_R** | 74.78 ± 0.17 | **GRCN_R** | 72.82 ± 1.03 | **SLAPS_R** | 77.12 ± 0.77 |
>
> **W 2: Random eges**.
> Thank you for this constructive suggestion. We add a baseline as the base model + random edges and conduct more experiments. The GCN+KNN and SLAPS methods are selected for experiments since they can work on all datasets. The number of random edges depends on the number of added edges in our methods. Following the experimental setting, we conduct five independent experiments and report the average accuracy along with the corresponding standard deviation as below. From the results, *we observe that adding random edges cannot guarantee performance improvement*. Therefore, we can make sure that the performance obtained by our methods is not just because there are more edges present.
>
> | **Datasets** | **ogbn-arxiv** | **Cora390** | **Cora140** | **Citeseer370** | **Citeseer120**  | **Pubmed** |
> |---|---|---|----|---|---|---|
> |**GCN+KNN**                    | 55.15 ± 0.11 | 72.82 ± 0.39 | 67.94 ± 0.29 | 73.28 ± 0.23 | 69.68 ± 0.53 | 68.66 ± 0.05 |
> |**GCN+KNN_RandEdge** | 54.31 ± 0.15 | 72.68 ± 0.13 | 67.76 ± 0.33 | 72.34 ± 0.29 | 68.86 ± 0.76 | 68.46 ± 0.16 |
> |**GCN+KNN_U (ours)**     | 55.82 ± 0.11 | 72.82 ± 0.21 | 68.18 ± 0.44 | 73.68 ± 0.10 | 69.74 ± 0.54 | 74.12 ± 0.32 |
> |**GCN+KNN_R (ours)**     | 55.86 ± 0.10 | 72.92 ± 0.28 | 68.12 ± 0.48 | 73.66 ± 0.14 | 69.90 ± 0.68 | 74.78 ± 0.17 |
> |**SLAPS**                          | 55.46 ± 0.12 | 76.62 ± 0.83 | 74.26 ± 0.53 | 74.32 ± 0.56 | 70.66 ± 0.97 | 74.86 ± 0.79 |
> |**SLAPS_RandEdge**       | 55.35 ± 0.50 | 76.68 ± 0.56 | 73.44 ± 0.67 | 74.08 ± 0.74 | 71.08 ± 0.81 | 74.10 ± 1.32 |
> |**SLAPS_U (ours)**           | 55.68 ± 0.09 | 76.94 ± 0.42 | 74.56 ± 0.21 | 74.82 ± 0.27 | 71.68 ± 0.47 | 76.74 ± 0.59 |
> |**SLAPS_R (ours)**           | 56.11 ± 0.15 | 76.82 ± 0.19 | 75.00 ± 0.49 | 74.90 ± 0.42 | 72.36 ± 0.49 | 77.12 ± 0.77 |
>
> **Q 1: M_U and M_R**.
> Thank you for pointing this out. In Line 268, M_U adopts the sparse intersection matrix $\hat U$ as the regularization, while M_Q combines $\hat U$ and $Q$ together. That means M_U refers to Equation 6 with a sparse $U$ (not $\widetilde U$ but $\hat U$, to solve the weight contribution rate decay issue), and M_R refers to Equation 8 with $\hat U$. We will make this statement more clear in the final version.
>
> **Q 2: Edge weights**.
> Yes. In fact, all the latent graph inference methods use real-valued edge weights from the output of their graph generators. The edge weights are calculated based on their designed distance functions. The smaller the distance between nodes, the larger the corresponding edge weights. For fairness and simplicity, we also use real-valued weights for the starved edges.
>
> **Q 3: Graph generators**.
> In the experiments, the type of graph generators depended on the baselines we used. For instance, in IDGL, the graph generator utilizes a weighted cosine similarity function as the metric function [4]. LCGS uses dual-normalization as the graph construction method [12]. For GRCN, we adopt a kNN graph [43]. And for SLAPS, we use the MLP-kNN generator, which corresponds to a multi-layer perceptron followed by the kNN operation [9].
>
> **L 1: Extend for higher values of k**.
> We thank the Reviewer for this valuable comment. As we stated in the supplementary, how to identify k-hop starved nodes for k>2 based on CUR decomposition remains an interesting direction for further investigation. Since we submitted this manuscript, we have been trying to solve this challenge. And now, we successfully come up with a solution. According to the definition of starved nodes, we know that a k-hop starved node also qualifies as a (k-1)-hop starved node. Therefore, we can iteratively identify k-hop starved nodes based on (k-1)-hop starved nodes. We use k=3 for illustration. Suppose that $rowmask$ is a $n$-dimensional mask vector where only the indexes of 2-hop starved nodes are marked as $1$ and $RM_+$ is the set of indexes of positive elements from $rowmask$. Then, we can obtain $R=A[rowmask, :]$. For each row $i$ of $R$, $V_i$ is a 3-hop starved node if, for all $j$ that satisfies $1_{\mathbb{R}^+}(R_{ij}=1)$, $j \in RM_+$. Based on the iterative strategy, we can solve the problem for higher values of ks.

---

> > ### Comment · Reviewer_tzHX · 2023-08-19
> > **Reviewer Response**
> >
> > Thanks for providing the new result and insights. Most of my concerns are now resolved and I have raised my soundness score and the overall rating accordingly.

---

> > > ### Author Response · Authors · 2023-08-19
> > > **Thank you very much for raising the score!**
> > >
> > > Dear Reviewer tzHX,
> > >
> > > We are very pleased that our responses resolved your concerns, and **thank you very much for raising the score!** We will try our best to revise our manuscript accordingly.
> > >
> > > Again, thank you so much for helping us improve the paper!
> > >
> > > Best regards,
> > > Authors

---

### Official Review · Reviewer_Qy2b · 2023-07-06

**Soundness:** 3 good
**Presentation:** 3 good
**Contribution:** 3 good
**Rating:** 7
**Confidence:** 4

**Summary:**

This paper proposes a new method for the latent graph inference problem. The motivation of the new method is the existence of supervision starvation nodes caused by graph sparsification operation. To reduce the number of supervision starvation nodes, the authors propose a CUR matrix decomposition based method to add an additional adjacency matrix to the original sparse adjacency matrix with some constraints. The extensive experimental results show that the proposed idea is effective on various datasets and different base models.

**Strengths:**

1.	The paper is well written. It is easy to follow the idea and understand the motivation and methodology design.
2.	Although the method is simple, it is effective and with sufficient analysis.
3.	The performance looks promising.


**Weaknesses:**

The reason why some nodes may receive no supervision is also because of the only one-layer GNN. If there are multiple layers of GNNs, the supervision information will propagate in the graph to achieve semi-supervised training, which is the core of most of GNNs. If there are multiple layers of GNNs, is that still reasonable to define ‘supervision starvation’? In Line 155, why is deeper GNN not an optimal solution to reduce the starving nodes? From the Figure 1, we can see that increasing GNN to 4 layers could reduce the starving nodes from near 3,000 to only near 500.

Since matrix A is dynamically generated from a latent graph generator, will the U or Q dynamically change as well? Are they dependent on generated A? If so, is that differentiable?



**Questions:**

see above

**Limitations:**

Yes

---

> ### Author Rebuttal · Authors · 2023-08-09
>
> **We really appreciate the Reviewer’s approval and constructive comments. We address your concerns as below.**
>
> **W 1**: Thank you for the constructive comments. We respond to this question in the following aspects:
> **Reason**. The reason why some nodes may receive no supervision is not because of the only one-layer GNNs. As stated in Line 105, Sec. 2.3, *graph sparsification is the main cause of the supervision starvation problem in LGI*. In other words, graph sparsification leads to some nodes being unable to receive supervision signals. Although increasing the number of layers of GNNs can alleviate this problem to some extent, starved nodes may still exist in multilayer GNNs, as shown in Figure 1.
> **Multilayer GNNs**. Yes, it is still reasonable to define "supervision starvation" if there are multiple layers of GNNs. Let us explain. To allow a k-hop starved node to receive information from labeled nodes, the GNNs need to have at least k+1 layers. However, as the number of layers increases, the number of nodes in each node’s receptive field grows exponentially, and the exponentially-growing amount of information needs to be squashed into a fixed-length vector. This will cause a so-called over-squashing issue [1], where crucial messages (supervision information here) fail to reach their distant destinations (starved nodes). That means, *as k increases, the GNNs may still fail to propagate supervision information from labeled nodes to k-hop starved nodes*. Therefore, we think it is still reasonable to define "supervision starvation" even if there are multiple layers of GNNs.
> **Optimal Solution**. Using deeper GNNs is not an optimal solution to the SS problem since it will bring up new issues, as illustrated below. **First**, although increasing GNN to 4 layers reduces the number of starved nodes from near 3,000 to near 500 for the Citeseer120 dataset, the number of starved nodes still accounts for nearly $15%$. In fact, the number of starved nodes depends on several factors, including the labeling rate of nodes, the graph sparsification process, and so on. *How to select the number of layers of GNNs to reduce the starved nodes for different datasets and different LGI methods is not easy*. **Second**, *the generalization of GNNs cannot be guaranteed by simply increasing the number of layers*. Deeper GNNs typically provide inferior performance due to over-smoothing [9][20] or over-squashing issues [1]. Empirically, we test the SLAPS method on the Pubmed dataset using a different number of layers, and the corresponding accuracies can be seen in the table below. **Moreover**, *as the number of layers increases, both the number of parameters and the float point operations (FLOPs) increase*. Following the above experiments, the number of parameters and the FLOPs for 2, 4, and 8-layer models are listed in the same table below. In fact, most LGI methods only adopt 2-layer GNNs for effectiveness and efficiency. We will add more discussion on this in our final version.
>
> | **Layers** | **Accuracy** | **Parameters** | **FLOPs** |
> |:--------:|:--------------:|:--------------:|:-------------:|
> | **2**| 74.86 ± 0.79 | 645.76K | 12.70G |
> | **4**| 70.24 ± 1.58 | 680.90K | 13.39G |
> | **8**| 41.18 ± 0.88 | 751.17K | 14.76G |
>
>
> **W 2:  U and Q**.
> Yes, U and Q will dynamically change as well. In fact, the values of U and Q can be obtained by different methods. For simplicity, we directly adopt the same strategy as in generating A. In this simple setting, U and Q depend on the generated A. Of course, they are both differentiable.

---

> > ### Comment · Reviewer_Qy2b · 2023-08-19
> > **Thanks for authors reply!**
> >
> > Thanks for the authors' reply!
> >
> > I am still unsure about the reasonability of "supervision starvation". Can we call all the semi-supervised learning tasks 'supervision starvation' because only a limited number of data points have correct label information?
> >
> > I will keep my score as before.

---

> > > ### Author Response · Authors · 2023-08-19
> > > **Thanks for your reply and question!**
> > >
> > > Dear Reviewer Qy2b,
> > >
> > > **Thank you very much for your reply and keeping a good score for us!**
> > >
> > > *No, we cannot call all the semi-supervised learning tasks “supervision starvation”*. The “supervision starvation” problem exists in semi-supervised latent graph inference tasks, not general semi-supervised tasks. Let us explain in detail.
> > >
> > > As we know, there are two types of weights in latent graph inference methods: **network weights** (i.e., network parameters in GNNs) and **edge weights** (the values of the adjacency matrix). *When we say “supervision starvation”, we mean that some edge weights (not network weights) receive no supervision from any labeled nodes through the semi-supervised training loss*. This will lead to poor generalization since **these under-trained edge weights are inevitably used to make predictions for testing samples at the test time**. In fact, existing latent graph inference methods commonly suffer from this problem.
> > >
> > > However, for general semi-supervised learning tasks (latent graph inference is not required), there is only one type of weight: **network weights** (i.e., network parameters). In this case, *all network weights can receive useful supervised information from a limited number of labeled data points through the semi-supervised training loss*. **And the predictions of training and testing samples are calculated based on the same network weights, which are well-trained**. Apparently, the “supervision starvation” problem does not exist here.
> > >
> > > In summary, *“**supervision starvation**” refers to some **edge weights** receiving no supervision in latent graph inference methods*. Therefore, we cannot call all the semi-supervised learning tasks “supervision starvation”.
> > >
> > > We hope this can address your concern. Thanks again for your help in improving our paper!
> > >
> > > Best regards,
> > >
> > > Authors

---

### Official Review · Reviewer_Qcor · 2023-07-31

**Soundness:** 3 good
**Presentation:** 2 fair
**Contribution:** 3 good
**Rating:** 5
**Confidence:** 3

**Summary:**

The paper introduces a model-agnostic enhancement for current LGI (latent graph inference) methods, aiming to address the assumed issue of unlearned features in a substantial number of nodes and edges, which are believed to negatively impact LGL's generalization performance. It proposes to learn a weighted residual graph, focusing on unlabeled nodes without labeled neighbors and intending to link them with some labeled nodes. Additionally, the proposed method applies a sparsity constraint to the residual graph, preventing significant alterations to the underlying graph topology learned from the base LGI model. Finally, the paper evaluates the effectiveness of this approach by integrating it with several prior LGI models, and finds modest improvements in standard graph learning tasks.

**Strengths:**

The paper's strength lies in its investigation of the important problem of optimizing the structure for neighborhood aggregation in graph neural networks. It introduces a method that addresses the issue of "supervision starvation" by adding labeled neighbors to initially supervision-starved nodes, which is demonstrated to bring benefit to existing LGI methods, especially on larger graphs.

**Weaknesses:**

1. The paper's proposed method shows moderate improvements, with limited statistical significance on small graphs (Cora and Citeseer). Larger graphs display more significant enhancements. The relationship between improvement and graph size should be explored for better insights. Further improvements for small graph scenarios are needed.

2. The paper lacks clarity in explaining the generation of the residual adjacency matrix. The emphasis on the relationship with CUR decomposition should be weaken since the learning criteria for weights in matrix $\tilde{U}$ are unrelated to reconstructing the target matrix $Q$. Furthermore, the method of selecting "supplementary adjacent points" from the labeled nodes for each 1-hop starved node remains unclear. The authors should be aware that "randomly selecting a subset of $\tau$ nodes from $c$ labeled nodes" is a different statistical event compared to "picking each labeled node at a probability of $\frac{\tau}{c}$". Improved explanations of these aspects are necessary to avoid confusion and strengthen the paper's presentation.

3. The paper lacks significant theoretical and technical novelty. Certain modeling choices are presented without adequate theoretical justification to demonstrate their importance or sensitivity analysis to show their insignificance. Specifically, the content in the paper should be enriched or reorganized to address the following research questions: (1) the reasons and mechanisms behind the significance of k-hop starved nodes in training a GNN through semi-supervised learning (i.e., why they are pivotal nodes), and (2) a comprehensive evaluation of the advantages and disadvantages of selecting k=1 to define the residual bipartite graph between k-hop starved nodes and labeled nodes, emphasizing how the benefits outweigh any drawbacks. Addressing these questions would enhance the paper's contributions and strengthen its overall impact in the field.

Given the aforementioned issues, a major revision of the current version of the paper is recommended before acceptance.

**Questions:**

1. Could the authors provide further elaboration on why the k-hop starved edges are defined as "having at least one endpoint being a $k$-hop starved node"? If the main objective is to identify edges whose weights are not trained in a $k$-layer GNN, wouldn't a more appropriate definition be "both endpoints are $(k-1)$-hop starved nodes"? (0-hop starved = unlabeled)

2. Could the authors provide an analysis of why adding connections between 1-hop starved nodes and labeled nodes improves the model's generalization? It appears that while introducing supervision signals to some unlabeled nodes, these additional connections to the labeled nodes may make them less "similar" to the unlabeled nodes. For example, the degree distribution of labeled and unlabeled nodes might be similar in the original dataset, but after the structural modifications proposed in the paper, the labeled nodes may exhibit significantly higher degrees than the unlabeled nodes. Wouldn't such divergence in the distribution of labeled and unlabeled nodes negatively impact the model's generalization? An explanation of these potential effects would help clarify the benefits and implications of the proposed structural modifications.

---

> ### Author Rebuttal · Authors · 2023-08-09
>
> **We thank the Reviewer for providing detailed comments. We address your concerns as below.**
>
> **W 1:  Graph size**.
> We would like to clarify that there is no direct relationship between improvement and graph size. We kindly remind the Reviewer that our method aims to eliminate starved nodes so as to utilize the missed supervision for better LGI. That means the improvement is directly related to the starved nodes, not the graph size. Besides, the number of starved nodes does not depend on the graph size but on the labeling rate of nodes and the graph sparsification process. As listed in Table 1, the results also do not show more enhancements on larger graphs and less improvement on smaller graphs. For example, SLAPS_R obtains a 1.7% improvement on Citeseer120 (nodes: 3,327, labeling rate: 3.61%) while only getting a 0.65% improvement on ogbn-arxiv (nodes: 169,343, labeling rate: 53.70%). Therefore, we think that exploring the relationship between improvement and graph size may not be sensible.
>
> **W 2: Generation of residual matrix**. The generation of matrix $B$ can be seen in Line 195-203, Sec 3.2. In fact, we do not generate $B$ directly. Instead, we generate a smaller matrix $\widetilde U$ first and then obtain $B$ by padding some rows and columns of zeros (see Eq. 6).
> **Matrix $Q$**. We would like to clarify that $Q$ is not the learning target, and we do not aim to reconstruct the matrix $Q$. $Q$ is only used in Definition 4 of “CUR Matrix Decomposition”. What we really want is the matrix $U$, which helps us identify and reduce the starved nodes.
> **Selection method**. As stated in Line 269, Sec 4, we select $\tau$ closest labeled nodes as the supplementary adjacent points. How to measure the distances depends on what baselines we used.
> **Probability**. We need to clarify that we do not “pick each labeled node at a probability of $\frac{\tau}{c}$”. Instead, we “randomly select $\tau$ nodes from $c$ labeled nodes” for each starved node. These two statements are totally different. In Proposition 1, we state “If there are $r$ 1-hop starved nodes, then for any $j \in CM_+$”, which means we consider the relationships between $r$ 1-hop starved nodes and one labeled node here. For any specific labeled node, say LN, if we randomly select $\tau$ nodes from $c$ labeled nodes for a 1-hop starved node, the probability of the node LN being selected is $\frac{\tau}{c}$. Obviously, this is totally different from the statement “pick each labeled node...” since we only consider one labeled node here.
>
> **W 3: Novelty**. In this paper, we identify that graph sparsification is the main cause of the SS problem, which is not explored by existing LGI methods. To address this issue, we propose to replenish the missed supervision for better LGI. Specifically, we first give a novel definition of starved nodes. Then, to identify the starved nodes, we design two simple solutions in Secs. 3.1 and 3.2 respectively. The starved nodes can be further diminished by our proposed regularization graphs. However, there still exists a potential issue we called weight contribution rate decay in Sec 3.3. That’s why we provide two modeling choices: M_U and M_R.
> **Reason**. k-hop starved noses are pivotal since we can tackle the SS problem directly by eliminating these nodes. In the SS problem, a number of edge weights are learned without any semantic supervision, which leads to poor generalization because the under-trained weights are used for the predictions of testing samples. If we eliminate the starved nodes, the missed supervision can be replenished to guide the learning and updating of these weights.
> **Definition of residual graph**. In fact, we can set any value for k to define residual graphs. However, if we set k>1, there will be a potential issue. For example, if we set k=3 and eliminate the 3-hop starved nodes, 1-hop and 2-hop starved nodes may still exist. To address this, a simple solution is to set k=1 to define the residual graph. Why only setting k=1 works? According to the definition of starved nodes, we know that a k-hop starved node also qualifies as a (k-1)-hop starved node. Therefore, if we eliminate 1-hop starved nodes then all m-hop starved nodes for m>1 will be eliminated, as stated in Line 203. Due to the effectiveness and simplicity, we select k=1.
>
> **Q 1:  Definition of starved edge**. Given a k-layer GNN, if one endpoint of an edge is a k-hop starved node, the weight of this edge is learned without any semantic supervision. This is because the k-hop neighbors of this node are all unlabeled, while a k-layer GNN can only aggregate signals from k-hop neighbors. According to this analysis, “both endpoints are (k-1)-hop starved nodes” is not a correct definition for k-hop starved edges. We kindly suggest the Reviewer check Definition 2 again for the meaning of k-hop starved nodes.
>
> **Q 2:  Generalization**. This question is essentially asking why eliminating starved nodes helps the model make better predictions for testing samples. As we stated in Line 100, Sec. 2.3, for a general LGI model, a number of edge weights are learned without any semantic supervision and cannot be semantically optimal after training. As a result, the model will exhibit poor generalization since the under-trained weights are used for the predictions of testing samples. Our methods replenish the missed supervision to guide the learning and updating of the under-trained weights, thus making better predictions. Besides, structural modification is necessary for LGI methods since we have absolutely no idea what the real structure of the data looks like. A good LGI model needs to infer a good latent structure from node features. The degree distribution is not a useful indicator that affects the generalization of a model. For example, when using full parameterization as the graph generator for SLAPS [9], all nodes have the same degrees. In this case, SLAPS can still obtain good or even better performance, as shown in [9].

---

> > ### Author Response · Authors · 2023-08-11
> > **Looking forward to further discussions**
> >
> > Dear Reviewer Qcor,
> >
> > Sorry to bother you! We are here to see if our responses have resolved your concerns. Do you expect us to have more analyses, results, or discussions for you to make a better evaluation of our paper? (If you do, please let us know asap so that we can have enough time to finish it.)
> >
> > Thank you very much again for your constructive comments.
> >
> > Best regards,
> > Authors of Sumbission 2239

---

> > > ### Author Response · Authors · 2023-08-17
> > > **Sincerely Expecting Further Discussion**
> > >
> > > Dear Reviewer Qcor,
> > >
> > > It is only 4 days away from the end (8/21) of the discussion period, and we haven't received any feedback from you regarding our responses. We are here to see if you could spend a few minutes checking our responses.
> > >
> > > Your concerns mainly relate to graph size, the generation of residual graphs, the definition of starved nodes, novelty, and generalization, for which we have dedicatedly provided more analysis and explanations to clarify our method. We do want to hear your further opinion about this, which is essential for us to improve the work. Thank you very much!
> > >
> > > Best regards,
> > > Authors

---

> > > > ### Comment · Reviewer_Qcor · 2023-08-20
> > > > **Thank you for the rebuttal. Here are some clarifications of my questions.**
> > > >
> > > > I thank the authors for providing a thorough explanation in response to my concerns. Following a careful review of the authors' rebuttal, I recognize the paper's importance in tackling the "supervision starvation" issue, which presents a novel research problem worthy of investigation. **Hence I decide to raise the score from 3 to 4**, and would like to keep updating the score if following questions can be discussed in revision:
> > > >
> > > > 1. About model generalization.
> > > >
> > > > The concept of "model generalization" usually pertains to how well a machine learning system performs on data it hasn't encountered before. However, in the context of transductive learning, there might be uncertainty regarding whether the test set can be truly considered "unseen" during the training phase. Despite this, my intention was to highlight to the authors that introducing connections between labeled and unlabeled nodes can lead to a substantial rise in the degrees of labeled nodes. This outcome might not always be advantageous, as it could cause the degree distribution of labeled nodes to differ from that of unlabeled nodes.
> > > >
> > > > My main skepticism revolves around the question of whether altering the distribution of node degrees will have an adverse effect on the model and to what extent. For instance, in the context of a node classification problem in a citation network, the degree of a node might be associated with the impact of the corresponding article. If there is a predictive connection between an article's impact and its subject label, the relationship between the degree and the label that the model learns from labeled nodes might not apply well to unlabeled nodes. This divergence in their degree distributions could potentially hinder the model's performance on unlabeled data.
> > > >
> > > > 2. About Graph size.
> > > >
> > > > While I concur with the authors regarding the non-influence of graph size on model performance, it's notable from Table 1 that the significance of performance enhancements is notably higher for PubMed and ogbn-arxiv compared to Cora and Citeseer. The distinction in benchmark groups includes graph size. While it's acknowledged that other potential variables are uncontrolled, we can't definitively conclude from Table 1 that graph size impacts model performance. I anticipate insights on "situations where the model excels" given that some improvements listed in Table 1 are less significant than others.
> > > >
> > > > 3. About the definition of $k$-hop starved weights.
> > > >
> > > > In lines125, 126,  the paper writes:
> > > > > According to the definition presented, it is evident that k-hop starved weights are precisely the ones that receive no semantic supervision from labels
> > > >
> > > > My understanding is that "$k$-hop starved weights" corresponds to "untrained weights in a $k$-layer GNN." To illustrate, consider a graph with 4 nodes: $u_1$, $u_2$, $u_3$, and $u_4$. The edges are $(u_1, u_2)$, $(u_2, u_3)$, and $(u_3, u_4)$. If nodes $u_1$ and $u_4$ are labeled, for a 1-layer GNN, the untrained weight is on edge $(u_2, u_3)$, hence $(u_2, u_3)$ is a 1-starved edge. However, according to Definition 2, none of the four nodes are considered 1-starved nodes. This situation appears perplexing, could the authors please provide clarification?
> > > >
> > > > 4. About the confusion in the selection method of sampling $\tau$ nodes from $c$ labeled nodes.
> > > >
> > > > The authors are correct, no revision is needed for this part.
> > > >
> > > > 5. About generation of the residual matrix.
> > > >
> > > > I recognize that the intention is not to generate a matrix for reconstruction purposes. As a result, I recommend considering a reduction in the association between the proposed method and CUR decomposition, which is primarily designed for matrix reconstruction. I personally don't find it necessary to introduce CUR decomposition to interpret the U matrix.

---

> > > > > ### Author Response · Authors · 2023-08-21
> > > > > **Thank you very much for raising the score!**
> > > > >
> > > > > Dear Reviewer Qcor,
> > > > >
> > > > > **Thank you so much for your reply and raising the score! We are very pleased that you recognize our paper’s importance**. After reading the clarifications carefully, we find that your questions are important and worth further exploring, which will improve our paper to a higher level. We address your questions below.
> > > > >
> > > > > Q1.
> > > > > Thanks for your further clarifications on this question. We concur with you that introducing connections between labeled and unlabeled nodes can lead to a substantial rise in the degrees of labeled nodes. It is actually a good question whether altering the distribution of node degrees will have an adverse effect on the model. To be honest, we have not yet observed any adverse effects from the current experimental results. We can provide a basic insight as to why introducing connections between labeled and unlabeled nodes helps the models. As we know, an unlabeled node must belong to a specific class, and in the node classification task, a specific class must have some labeled nodes. As a result, we can derive the prior assumption that a good classification model will make an unlabeled node closer to some labeled nodes of the same class. That’s why we select the closest labeled nodes as the supplementary adjacent points for the starved nodes. If we compulsorily add such connections between the unlabeled node and its closest labeled nodes, the prior information will be exploited during training, helping the models work better.
> > > > > Nevertheless, it is still worth exploring whether the divergence in degree distributions will potentially hinder the model’s performance. What the exact relationship is between the degree distributions and the model’s performance is a difficult and open question that deserves in-depth exploration. We think the reviewer provides us with a good potential direction to research deeper. Due to time limitations, we will explore this in the future.
> > > > >
> > > > > Q2.
> > > > > We are pleased that you agree with us regarding the non-influence of graph size on model performance. For the situations where our model excels, we provide some basic insights as follows. First, we think the labeling rate of nodes is the key factor that affects the significance of improvement. As we can see from Table 1, PubMed has the lowest labeling rate, while the improvement is the most significant. Second, the improvement also depends on the baselines we used. Different baselines have different graph inference strategies and graph sparsification processes, which also makes the improvement significance different. For example, we can observe that our methods obtain more than 1% improvement for the GCN&KNN model on the Cora and CiteSeer datasets, while the improvement for the GCN+KNN model is not significant. In summary, several potential factors (e.g., labeling rate, graph sparsification process, datasets, baselines, etc.) may affect the improvement of our models. We only provide some basic insights on this since it is actually difficult to say exactly which situation our method is definitely good in.
> > > > >
> > > > > Q3.
> > > > > Thanks for pointing this out and providing an illustration. We have carefully read your clarifications, and we finally recognize that your understanding is correct. Note that a k-hop starved node also qualifies as a (k-1)-hop starved node. It is actually more appropriate to define the k-hop starved edges as “both endpoints are (k-1)-hop starved nodes (0-hop starved = unlabeled)”. We will update the definition accordingly in the revision. Thanks again for your kind reminder and suggestion!
> > > > >
> > > > > Q4.
> > > > > We are very pleased that our responses can successfully address your concern on this point. Thanks for the time spent reading the detailed responses!
> > > > >
> > > > > Q5.
> > > > > Due to space limitations, we had to delete some explanations on the target matrix Q at the first rebuttal. Now, let us discuss this more. In fact, our initial intention was to construct the target matrix Q (in Eq. 5) and use it as the regularization matrix B (in Eq. 4). It is, of course, sensible and feasible. However, we found a potential drawback to this design. That is, the reconstruction of target Q requires matrix multiplications of three matrices, which is very time-consuming. To tackle this problem, we unexpectedly found that only constructing matrix U is enough to solve the supervision starvation problem. That’s the reason why we emphasize and focus on matrix U in the manuscript.
> > > > > We recognize that your suggestion is also reasonable. To make a better association between the proposed method and CUR decomposition, we will discuss the above initial design in the revision, add a variant that uses the target Q as the regularization, and provide more experimental results. In comparison with reduction, we think this is a better way to make our paper comprehensive and consistent with a better logical flow.
> > > > >
> > > > > **We will discuss these questions in the final revision. We sincerely thank the reviewer for helping us improve the paper!**
> > > > >
> > > > > Best regards,
> > > > > Authors

---

> > > > > > ### Comment · Reviewer_Qcor · 2023-08-21
> > > > > > **Follow-ups**
> > > > > >
> > > > > > 1. I am very interested in the response to Q5 in the above comment. Could the authors confirm for me that there are two variants of $\Gamma(\tilde{\mathbf{U}}, n)$, one is padding 0 into $\tilde{\mathbf{U}}$ as introduced in the paper, another is $\Gamma(\tilde{\mathbf{U}}, n) = \mathbf{C}\tilde{\mathbf{U}}\mathbf{R}$, and the first variant is more efficient than the second variant?
> > > > > >
> > > > > > 2. I'm pleased that I grasped the notion of $k$-starved weights accurately. I recommend to the authors that they consider adding a supplementary note clarifying that self-connections aren't included within the definition of any neighborhood. Such clarification would prevent potential confusion concerning nodes classified as $0$-starved and $1$-starved.
> > > > > >
> > > > > > 3. As a follow-up on the discussions about model generalization, I find the authors' explanations to be persuasive. Within the context of transductive learning, the discrepancy between labeled and unlabeled nodes might carry less weight compared to the challenge of supervision scarcity. Nonetheless, this aspect could present a possible concern for inductive learning scenarios. While I comprehend that delving into this matter goes beyond the paper's current scope, it would certainly be valuable to include it in future discussions.

---

> > > > > > > ### Author Response · Authors · 2023-08-21
> > > > > > > **Thank you for your reply and further questions!**
> > > > > > >
> > > > > > > Dear Reviewer Qcor,
> > > > > > >
> > > > > > > Thank you very much for your reply and further questions! We address your concerns below.
> > > > > > >
> > > > > > > Q1.
> > > > > > > Yes, you are right. There are two variants, and the first variant (padding 0) is more efficient than the second variant. We can also provide experimental results to demonstrate this point. Due to time limitations, we will add the results to the final revision.
> > > > > > >
> > > > > > > Q2.
> > > > > > > Thank you very much for this suggestion. It is sensible to make such a clarification. We will add a supplementary note to clarify that self-connections aren’t included within the definition of any neighborhood. Thanks again for pointing this important information out for us!
> > > > > > >
> > > > > > > Q3.
> > > > > > > Thanks for your agreement with our explanations. We totally concur with the reviewer that it is valuable to include this matter in future discussions. We thank the reviewer for this important suggestion, and we will definitely discuss this point in our final revision.
> > > > > > >
> > > > > > > **Thanks again for providing us with so many important suggestions and new insights.**
> > > > > > >
> > > > > > > Best regards,
> > > > > > > Authors

---

> > > > > > > > ### Comment · Reviewer_Qcor · 2023-08-21
> > > > > > > > **Upscore decision**
> > > > > > > >
> > > > > > > > The paper's notable strength lies in its ability to demonstrate to the community that tackling the issue of supervision scarcity holds substantial promise as a direction for advancing latent graph learning methodologies. While certain concerns related to presentation clarity were initially raised, the authors have committed to making necessary enhancements, which have been duly addressed. Personally, I believe that encouraging papers that introduce intriguing topics to the community is essential, and thus, I am biased towards recommending the acceptance of this paper. Wishing the authors the best of luck.

---

> > > > > > > > > ### Author Response · Authors · 2023-08-21
> > > > > > > > > **Thank you very much for recommending the acceptance of our paper!**
> > > > > > > > >
> > > > > > > > > Dear Reviewer Qcor,
> > > > > > > > >
> > > > > > > > > **Thank you very much for recommending the acceptance of our paper**.
> > > > > > > > >
> > > > > > > > > We are very pleased that you can recognize our contributions and provide us with valuable suggestions and new insights! We will try our best to revise the manuscript accordingly. Again, many thanks for helping us improve the paper!
> > > > > > > > >
> > > > > > > > > Best regards,
> > > > > > > > > Authors

---

### Decision · Program_Chairs · 2023-09-21

**Decision:**

Accept (poster)

**Comment:**

After discussions with the authors, all five reviews ultimately leaned on the side of acceptance, although two of the five reviewers consired the work only weakly or borderline acceptable.

Several aspects of the work were appreciated:

+ The problem of optimizing the neighborhood aggregation structure in graph neural networks was considered important (Qcor,tzHX,se4r).

+ The supervision starvation problem was considered an important direction (tzHx,6cu2).

+ The method was considered well-motivated and its model-agnostic nature was appreciated (6cu2)

+ The method was considered effective (Qy2b,tzHX) with promising performance (Qy2b,6cu2)

+ The analysis was considered solid (se4r) and sufficient (Qy2b)

+ The theoretical support was appreciated (se4r,6cu2)

+ The paper was considered well written by some reviewers (Qy2b,se4r,6cu2)

However, multiple concerns were also raised:

- The performance improvement was considered only moderate (Qcor) and more investigation of relationship to graph size and improvements for small graphs were desired (Qcor)

- The paper was considered to lack theoretical and technical novelty (Qcor)

- Better theoretical justification of some modeling choices and their sensitivity was desired (Qcor)

- Using average of top-5 testing performances (not testing performance of models having top validation performance) was considered problematic (tzHX)

- Comparison to state of the art GNNs was desired (se4r). Authors provided some results in a rebuttal.

- Comparing only to a base model without any added edges was considered problematic, and comparison to make sure performance is not just due to more edges was desired (tzHX). Authors provided some new results on this in a rebuttal.

- Lack of clarity in some technical details was criticized (Qcor), and some sections were considered hard to follow (se4r)

- Some technical questions regarding reasonability of supervision starvation and its relationship to semisupervised learning, especially in multi-layer GNNs, remained (Gq2b)

- There was concern about sensitivity to hyperparameters and performance on unweighted graphs (6cu2). Authors provided some discussion and results in a rebuttal.

- More explanation of efficiency improvements was desired (Se4r). Authors provided some additional results in a rebuttal.
Despite discussion with authors, concerns regarding generalization (Qcor), effect of graph size (Qcor), reasonability of the supervision starvation concept (Qy2b), comparison to state of the art GNNs (se4r), and some technical details (Qcor) seemed to partly remain.

Overall, although some concerns still remain, it seems there may be enough merits to the work that it could be presented at NeurIPS.